# Epoxy-oxylipins direct monocyte fate in inflammatory resolution in humans

Olivia V. Bracken[1], Parinaaz Jalali[1], James R. W. Glanville[1], Larrissa Benvenutti[1], Emma S. Chambers [2], Hugh Trahair[1], Madhur Motwani[1], Karen T. Feehan [1,3], Jamie G. Evans[1], Jhonatan de Souza Carvalho[1], Roel P. H. De Maeyer [1,4], Arne N. Akbar [1], Fred B. Lih [5], Darryl C. Zeldin[5], David Bishop-Bailey [1] ✉, Matthew L. Edin[5] & Derek W. Gilroy [1] ✉

The role of cytochrome P450-derived epoxy-oxylipins and their metabolites in human inflammation and resolution is unknown. We report that epoxy-oxylipins are present in blood of healthy, male volunteers at baseline and following intradermal injection of UV-killed *Escherichia coli*, an experimental model of acute resolving inflammation. At the site of inflammation, cytochrome P450s and epoxide hydrolase (EH) isoforms, which catabolise oxylipins to corresponding diols, are differentially upregulated throughout the inflammatory response, as is the biosynthesis of epoxy-oxylipins. GSK2256294, a selective sEH inhibitor specifically elevates 12,13-EpOME and 14,15-EET. While inhibition of sEH hastens pain resolution, it has no effect on tissue heat, redness and swelling. GSK2256294, however, significantly reduces numbers of circulating intermediate monocytes that expand during inflammation. We find that 12,13-EpOME blocks the transition of classical to intermediate monocytes in a p38 MAPK-dependent manner, results that are recapitulated when blocking p38 MAPK in vitro and when administering the p38 MAPK inhibitor losmapimod in vivo to healthy volunteers. Furthermore, fewer intermediate monocytes are observed at the site of inflammation, accompanied by reduced tissue CD4 T cells. Hence, we have mapped the expression, activity and function of epoxy-oxylipins in human inflammation revealing new mechanisms of monocyte differentiation and resolution biology.

There is much to learn about the regulatory mechanisms that control inflammatory responses as well as those that bring about their resolution. Gaining a better understanding of these protective pathways and how they are dysregulated by factors such as genetic abnormalities[1,2], ageing[3] or the environment[4], will inform on why inflammation can change from being our greatest ally against infection and injury, to a serious threat as a result of failed resolution resulting in chronicity of disease.

While many soluble mediators and cellular processes necessary for effective resolution have been described[5], the lack of curative therapeutics calls for more research into understanding the complexity of inflammatory responses. One under investigated pathway in

[1]Department of Ageing, Rheumatology and Regenerative Medicine, Division of Medicine, University College London, London, UK. [2]Centre for Immunobiology, Blizard Institute, Queen Mary University of London, London, UK. [3]School of Cancer and Pharmaceutical Sciences, King's College London, London, UK. [4]Nuffield Department of Orthopaedics, Rheumatology and Musculoskeletal Sciences, Botnar Research Centre, University of Oxford, Oxford, UK. [5]Division of Intramural Research, National Institute of Environmental Health Sciences, National Institutes of Health, Research Triangle Park, Durham, NC, USA. ✉e-mail: bishopbaileyd@gmail.com; d.gilroy@ucl.ac.uk

human immunology is that of the cytochrome P450 (CYP450) epoxygenases. The CYP450 epoxygenases convert polyunsaturated fatty acids (PUFAs) into epoxy-oxylipins. Linoleic acid (LA) is metabolised into epoxy-octadecenoic acids (EpOMEs), arachidonic acid (AA) into epoxy-eicosatetraenoic acids (EETs), docosahexaenoic acid (DHA) into epoxy-docosapentanoic acids (EpDPA/EpDPE) and eicosapentaenoic acid (EPA) into epoxy-eicosatetraenoic acids (EpETEs)[6,7]. In humans,

there are 41 protein-coding subfamilies comprising 57 genes, while the murine genome contains 102 full-length putatively functional genes[8]. The CYP450 enzymes most associated with epoxy-oxylipin production in humans are from the CYP2C family, namely CYP2C8 and CYP2C9, and CYP2J2[9]. After production, the epoxy-oxylipins are catabolised into more stable diols by a group of enzymes known as the epoxide hydrolases (EH). The primary epoxide hydrolase enzymes are

**Fig. 1 | Expression of CYP2J2 and sEH and lipid profiles during inflammation in human skin tissue.** Paraffin-embedded skin punch biopsies were obtained at baseline, 4, 8 and 24 h post UVKEc injection. Sections were incubated with **A** CYP2J2 and **B** sEH primary antibodies and visualised by avidin-biotin complex-based detection methods. The sections were subsequently counterstained with haematoxylin. **C** Skin biopsies at baseline, 4, 8 and 24 h were stained and the % area of staining quantified for (i) CYP2J2, (ii) Pan CYP2C, (iii) sEH, (iv) mEH ($n = 1–3$; biologically independent samples). **D** Healthy volunteers received an intradermal injection into the forearm of UV-killed *E. coli* (UV-KEc), resulting in a local inflammatory response. Local inflammatory exudate was collected at baseline, 4, 8, 14, 24, 48 and 72 h post UVKEc injection and analysed by mass spectrometry for lipidomic profiles. (i) Levels of polyunsaturated fatty acids (PUFAs) were quantified in pg/ml ($n = 2–5$; biologically independent samples). Cytochrome P450 products and Epoxide Hydrolase products (ii) 14,15-DHET, (iii) 12,13-DiHOME, (iv) 19,20-DiHDPA and (v) 17,18-DiHETE were quantified during inflammation in pg/ml ($n = 4$-5; biologically independent samples). (vi) 12,13-EpOME was quantified in pg/ml ($n = 4$-5; biologically independent samples). The dotted line represents baseline levels of the lipids. (vii) Total CYP450 profiles (viii) total COX products were normalised relative to baseline levels to show the changing profile during acute and resolving inflammation (each dot is representative of 4–5 biologically independent samples). Data were assessed for normalisation using the D'Agostino & Pearson test, the Shapiro–Wilk test and visualised using a QQ plot. Data are presented as mean ± SD. Source data are provided as a Source Data file.

microsomal EH (mEH) and soluble epoxide hydrolase (sEH), although a further two have been identified in the human genome: Epoxide Hydrolase 3 (EH3) and Epoxide Hydrolase 4 (EH4)[10–12]. mEH is a major metaboliser of xenobiotic compounds and, thus its therapeutic inhibition is likely to result in side effects[13]. However, inhibiting sEH is proven safe and effective[14]. Hence, inhibiting sEH elevates levels of epoxy-oxylipins and represents a useful tool to understand their role in health and disease.

The immuno-modulatory properties of the epoxy-oxylipins include inhibiting NF-κB and blocking p38 MAPK phosphorylation, resulting in reduced expression of pro-inflammatory cytokines and cell adhesion molecules[15,16]. Epoxy-oxylipins have also been shown to induce angiogenesis via the production of VEGF[17], while in murine models, we have shown that sEH depletion enhances resolution of acute peritonitis[18]. There is a large body of evidence from others showing sEH inhibition reduces inflammation in mouse models of Alzheimer's disease[19], diabetic retinopathy[20], sepsis[21] and autoimmunity[22]; a body of literature also exists on the analgesic effects of the epoxy-oxylipins both in rodents and in larger mammals, including horses and dogs[23–26]. However, little is known about how epoxy-oxylipins impact inflammation and resolution in humans.

GSK2256294 is a potent and reversible inhibitor of sEH[27–30]. It is safe and well tolerated in humans with maximum systemic concentrations achieved within 1–2 h of dosing, with the average inhibition of sEH being 99.8% and a half-life of 20–30 h[31]. Using GSK2256294 in our model of UV-killed *E. coli* (UV-KEc)-induced dermal inflammation[32–35], we report on the epoxy-oxylipins elevated following sEH inhibition and how they modulate the salient, molecular and cellular aspects of local and systemic inflammation and its resolution.

In this work, we show that sEH inhibition significantly elevates both plasma and tissue levels of the LA-derived epoxy-oxylipin, 12,13-EpOME, which results in a significant reduction of intermediate monocytes during inflammation, both in peripheral blood and in the skin. Furthermore, we show that 12,13-EpOME significantly inhibits the phosphorylation of p38 MAPK, which, when inhibited in vitro and in vivo using losmapimod, an oral inhibitor for p38 MAPK, significantly reduces the presence of intermediate monocytes.

## Results

### CYP/EH isoform profiles and lipid metabolites in resolution

The expression of key enzymes in the synthesis and metabolism of epoxy-oxylipins, namely CYP2J2, the CYP2C family and the epoxide hydrolase enzymes, was characterised in UVKEc-elicited dermal inflammation. Under baseline conditions, CYP2J2 was found in the epidermis, hair follicles and sweat glands, but not in the reticular dermis (Fig. 1A). Following injection of UVKEc, CYP2J2 was observed in the infiltrating immune cells at 4 and 24 h, primarily in the reticular dermis (Fig. 1A, C). A similar profile of expression was found for the CYP2C family of enzymes, Supplementary Fig. 1 and Fig. 1C.

sEH was detected in the epidermis, hair follicles and sweat glands at baseline (Fig. 1B). As expected, sEH increased following onset of inflammation and was maintained at twice the level of baseline up to 24 h (Fig. 1B, C). mEH was found within the epidermis and dermis at all time points with the highest level of expression found in the papillary layer (Supplementary Fig. 1). Levels of mEH remained unchanged from baseline to the early phase of inflammation (8 h) but increased ×1.6 fold at 24 h (Fig. 1C). In contrast, EH3 was expressed in the epidermis, hair follicles and sweat glands and decreased during acute inflammation (4 and 8 h post intradermal UV-KEc injection), returning to baseline by 24 h (Supplementary Fig. 1).

Using negative pressure to elicit a blister over the site of inflammation to collect the inflammatory exudate, we quantified the levels of the PUFAs AA, LA and DHA, as well as their respective metabolites by sEH, namely 14,15-DHET, 12,13-DiHOME, 19,20-DiHDPA during inflammation and its resolution (Fig. 1D). Additionally, we quantified the LA CYP450 metabolite, 12,13-EpOME (Fig. 1D). It is worth noting that the levels of 12,13-DiHOME are nearly 100 times higher at 14 h compared to 14,15-DHET and 19,20-DiHDPA. Although we could not detect the PUFA EPA, we were able to detect its sEH metabolite 17,18-DiHETE. The CYP450 products peak at 14 h post UV-KEc injection, implying they play a role in the resolution of inflammation, whereas the COX products peak at 4 h, during the acute phase of the inflammatory response (Fig. 1D).

### Effect of sEH inhibition on the core features of inflammation

Given that epoxy-oxylipins were elevated in inflammatory exudates at the transition from onset to resolution (Fig. 1), we felt that a prophylactic and a separate therapeutic inhibition of sEH was needed to distinguish constitutive epoxy-oxylipins from those elicited by intradermal UVKEc. Therefore, we followed the prophylactic and therapeutic dosing regimens outlined in Fig. 2A, B. A table of characteristics for the participants recruited into this study for the prophylactic and therapeutic arms can be found in Tables 1 and 2, respectively. In the prophylactic arm of the study, participants were dosed with 15 mg GSK2256294 2 h prior to the onset of inflammation. In the therapeutic arm of the study, participants were dosed with 15 mg GSK2256294 4 h after the injection of UV-KEc.

Perception of pain peaked between 4 and 24 h in this model, with GSK2256294 having no analgesic effect during this phase, Fig. 2C, D. However, at 24 h in prophylactic participants and between 24 and 48 h in therapeutic participants, the resolution of pain was hastened with sEH inhibition (Fig. 2C, D). Local temperature peaked at 24 h with inhibition of sEH having no effect on this inflammatory parameter (Fig. 2C, D). Furthermore, GSK2256294 did not affect oedema, measured indirectly by quantifying blister fluid volume (Fig. 2C, D). Injection of UVKEc results in increased microvascular reactivity quantified by laser doppler analysis, peaking at 24 h (Fig. 2E), although inhibition of sEH did not overtly affect local redness (Fig. 2E).

Taken together, except for pain, inhibition of sEH with GSK2256294 did not significantly alter the clinical features associated with UVKEc injection.

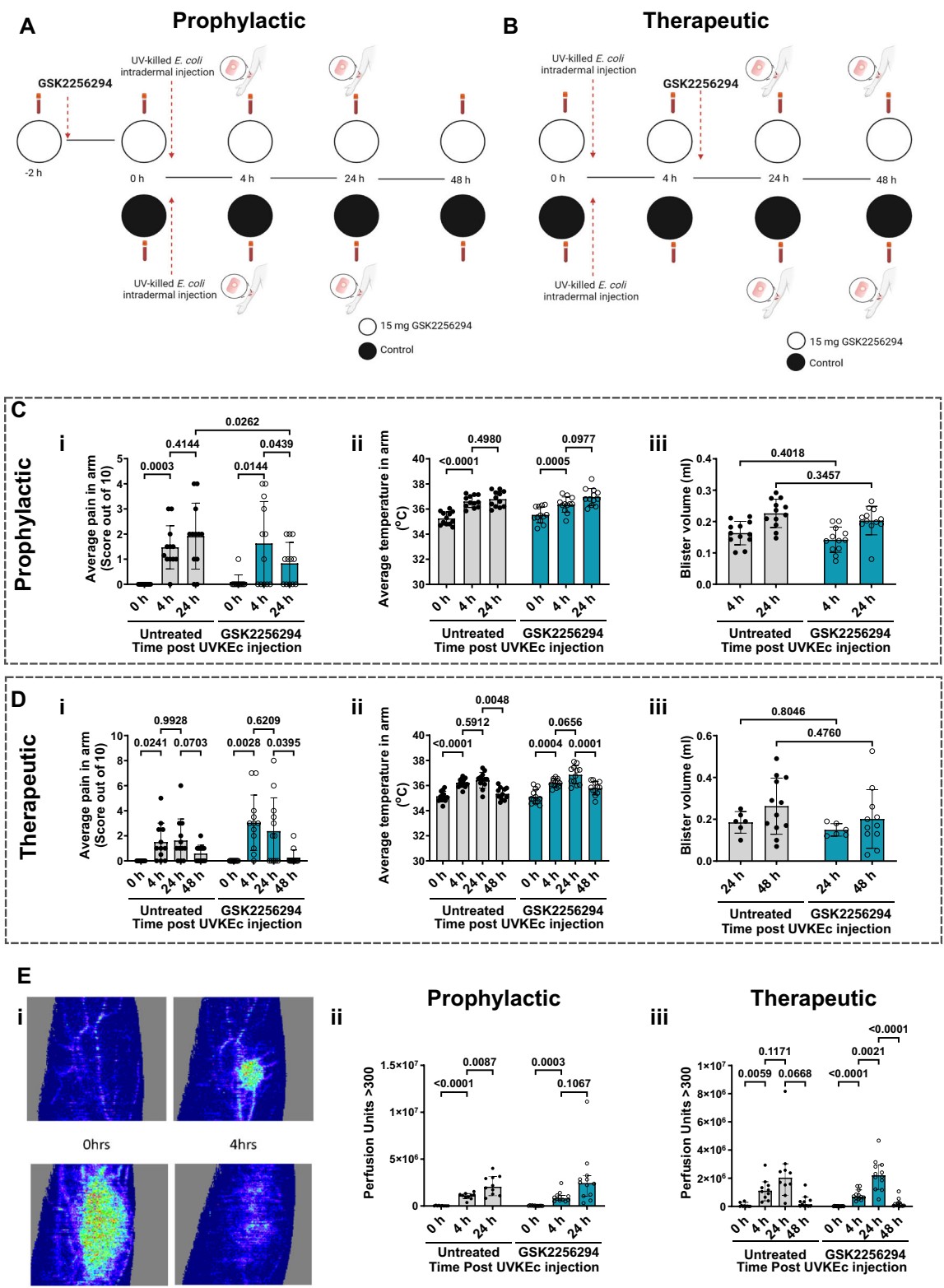

## Effect of sEH inhibition on plasma epoxy-oxylipins

To determine which epoxy-oxylipins were altered following sEH inhibition, we performed detailed lipidomic analysis. Plasma levels of the four PUFAs from which the epoxy-oxylipins are derived are unchanged with the induction of inflammation and, as expected, are unaffected by dosing participants with GSK2256294 (Supplementary Fig. 2).

Following the pre-dosing approach described in Fig. 2A, GSK2256294 significantly elevated plasma 14,15-EET and 12,13-EpOME within 2 h, which resulted in an increase in the ratio of 12,13-EpOME:12,13-DiHOME and 14:15-EET:14,15-DHET (Fig. 3A), a reliable index of sEH inhibition. Following induction of inflammation, prophylactic sEH inhibition resulted in a significant increase in the levels of 14,15-EET and 12,13-EpOME, concomitant with a significant increase in the ratio of 14,15-EET:14,15-DHET and 12,13-EpOME:12,13-DiHOME at 4, 24 and 48 h (Fig. 3B). In the therapeutic dosing regime, GSK2256294 caused an increase in 14,15-EET at 24 and 48 h and a corresponding

**Fig. 2 | Study design with GSK2256294 for prophylactic and therapeutic dosing regimens and clinical score data. A** Prophylactic study design. Participants received GSK2256294 2 h before inflammation was induced by intradermal UVKEc injection. Blood samples were taken before dosing, 2 h after dosing, and at 4, 24 and 48 h post-inflammation. Blisters were raised at 4 and 24 h. **B** Therapeutic study design. Participants received GSK2256294 4 h after inflammation was induced by intradermal UVKEc injection. Blood was collected at baseline and at 4, 24 and 48 h post-inflammation. Blisters were raised at 24 and 48 h. **C** Clinical scores in the prophylactic arm of the study (i) average pain score scored from 0 to 10 in each arm; (ii) average temperature at the inflammatory site in degrees Celsius (°C); (iii) blister volume in ml (untreated: $n = 12$; GSK2256294: $n = 12$; biologically independent samples). **D** Clinical scores in the therapeutic arm of the study (i) average pain score scored from 0 to 10 in each arm; (ii) average temperature at the inflammatory site in °C; (iii) blister volume in ml (untreated: $n = 12$; GSK2256294: $n = 6–11$;

biologically independent samples). Data for **C**, **D** are presented as mean ± SD. **E** Inflammatory sites were imaged at baseline, 4, 24 and 48 h after UVKEc injection using laser doppler imaging. Blood flow was quantified as perfusion units (flux × valid pixels) with a threshold of >300 flux units. (i) Representative doppler images of blood flow. (ii) Perfusion units with prophylactic dosing of GSK2256294 (untreated: $n = 10$; GSK2256294: $n = 12$; biologically independent samples). (iii) Perfusion units with therapeutic dosing of GSK2256294 (untreated: $n = 10$-11; GSK2256294: $n = 12$; biologically independent samples). Data were assessed for normalisation using the D'Agostino & Pearson test, the Shapiro−Wilk test and visualised using a QQ plot. For **C**(i−ii), **D**(i−ii), and **E**(ii−iii), data were analysed using two-way ANOVA mixed-effects models with Tukey's multiple-comparison test. **C**(iii) and **D**(iii) were analysed using two-way ANOVA mixed-effects models with Šídák's correction. Source data are provided as a Source Data file. *Created in BioRender. Bracken, O. (2026)* https://BioRender.com/xvt2mut.

increase in the ratio of 14,15-EET:14,15-DHET (Fig. 3C). The ratio of 12,13-EpOME:12,13-DiHOME was also significantly increased at 24 and 48 h following therapeutic dosing compared to untreated participants (Fig. 3C). Although we focused here on 12,13-EpOME and 14,15-EET, we quantified all other CYP450 derived lipids, a table of which can be found in Supplementary Fig. 3.

Hence, these data demonstrate efficient inhibition of sEH by GSK2256294 in this model.

### Inhibiting sEH blocks the expansion of intermediate monocytes

Employing an antibody panel designed to capture changes in all blood immune cells, using manual gating in FlowJo, we noted that GSK2256294 had a striking effect on blood monocyte populations. Specifically, in both interventional arms of the study, sEH inhibition blocked the expansion of intermediate and non-classical monocytes between 4 and 24 h (Fig. 4A); with a significant difference in the number of intermediate monocytes at 24 h between treated and untreated groups in the prophylactic arm. The gating strategy for the monocyte subsets can be found in Supplementary Fig. 4. These data show, unexpectedly, that sEH inhibition blocks the expansion of

**Table 1 | Characteristics of volunteers recruited into the prophylactic arm of the study**

| Treatment group | Untreated | 15 mg GSK2256294 |
|---|---|---|
| Mean age | 25.3 | 25.9 |
| Age SD | 4.1 | 4.1 |
| Recruited | 12 | 12 |
| Excluded | 0 | 0 |

24 healthy, male volunteers between the age of 18–50 were recruited into the prophylactic arm of this study. All participants were subject to screening to assess eligibility and to consent to participation.

**Table 2 | Characteristics of volunteers recruited into the therapeutic arm of the study**

| Treatment group | Untreated | 15 mg GSK2256294 |
|---|---|---|
| Mean age | 24.16 | 23.16 |
| Age SD | 3.9 | 2.6 |
| Recruited | 12 | 13 |
| Excluded | 0 | 1 |
| 48 h only | 5/12 | 5/12 |

25 healthy, male volunteers between the age of 18–50 were recruited into the prophylactic arm of this study. All participants were subject to screening to assess eligibility and to consent to participation. One participant was excluded from this study due to failing the eligibility requirements.

intermediate and non-classical monocytes in humans during inflammation.

Data were also processed by unbiased multiparametric analysis using the CATALYST package in *R* where we identified six groups of monocytes−two sub-categories of classical monocytes (CM (1) and CM (2)), one population that is transitioning between classical-intermediate monocytes (CM-IM), one distinct intermediate population (IM), a population transitioning from intermediate to non-classical monocytes (IM-NCM) and, finally, nonclassical monocytes (NCM); all presented as Uniform Manifold Approximation and Projection (UMAP) (Fig. 4B) with the gating strategy for the monocyte input shown in Supplementary Fig. 4. Expression markers charted the differentiation of these populations with CD14, CCR2, CD64 and CD205 typically declining as classical monocytes differentiate with a corresponding increase in CD16, HLA-DR, CD86 and CD11c as cells acquire a non-classical phenotype[33], (Fig. 4B). These data support analysis using manual gating (Fig. 4A and Supplementary Fig. 4)−that GSK2256294 inhibits the expansion of intermediate monocytes between 4 and 24 h following UVKEc injection (Fig. 4B). To exclude margination on the microvascular endothelium as an explanation for reduced intermediate numbers, we quantified expression of ICAM-1, PSGL-1, VCAM-1, CCR2 and CX3CR1 on these cells and found no difference between the control and GSK2256294 treated group (Fig. 4C). Therefore, given the current paradigm of CM-IM-NCM linear differentiation[33,36,37], we suggest that sEH inhibition blocks the expansion of intermediate monocytes in blood of volunteers injected with intradermal UV-KEc. For the effects of GSK2256294 on other peripheral blood cell numbers, see Supplementary Fig. 5.

### 12,13-EpOME blocks monocyte differentiation by inhibiting p38 MAPK

To understand how sEH inhibition blocks the expansion of intermediate monocytes in blood, we incubated 14,15-EET and 12,13-EpOME with PBMCs and monitored monocyte differentiation in vitro. 12,13-EpOME, but not 12,13-DiHOME, partially blocked classical to intermediate monocyte differentiation with both 14,15-EET and 14,15-DHET being without effect (Fig. 5A and Supplementary Fig. 6). Representative flow plots from these experiments can be found in Supplementary Fig. 7. Investigating the molecular basis of this inhibition, it is reported that 12,13-EpOME blocks the phosphorylation of p38 MAPK[15]. Phospho-p38 MAPK was enriched in classical monocytes, with phosphorylation levels declining as monocytes differentiate towards the non-classical phenotype (Fig. 5B and Supplementary Fig. 8). 12,13-EpOME significantly blocked phosphorylation of p38 MAPK, in monocytes (Fig. 5C). By corollary, losmapimod, the selective p38 MAPK inhibitor, also blocked monocyte differentiation in vitro (Fig. 5D). Furthermore, participants dosed orally with losmapimod for four days resulted in a significant decrease in the percent of intermediate monocytes in peripheral blood (Fig. 5E). Collectively, these data suggest that 12,13-EpOME and

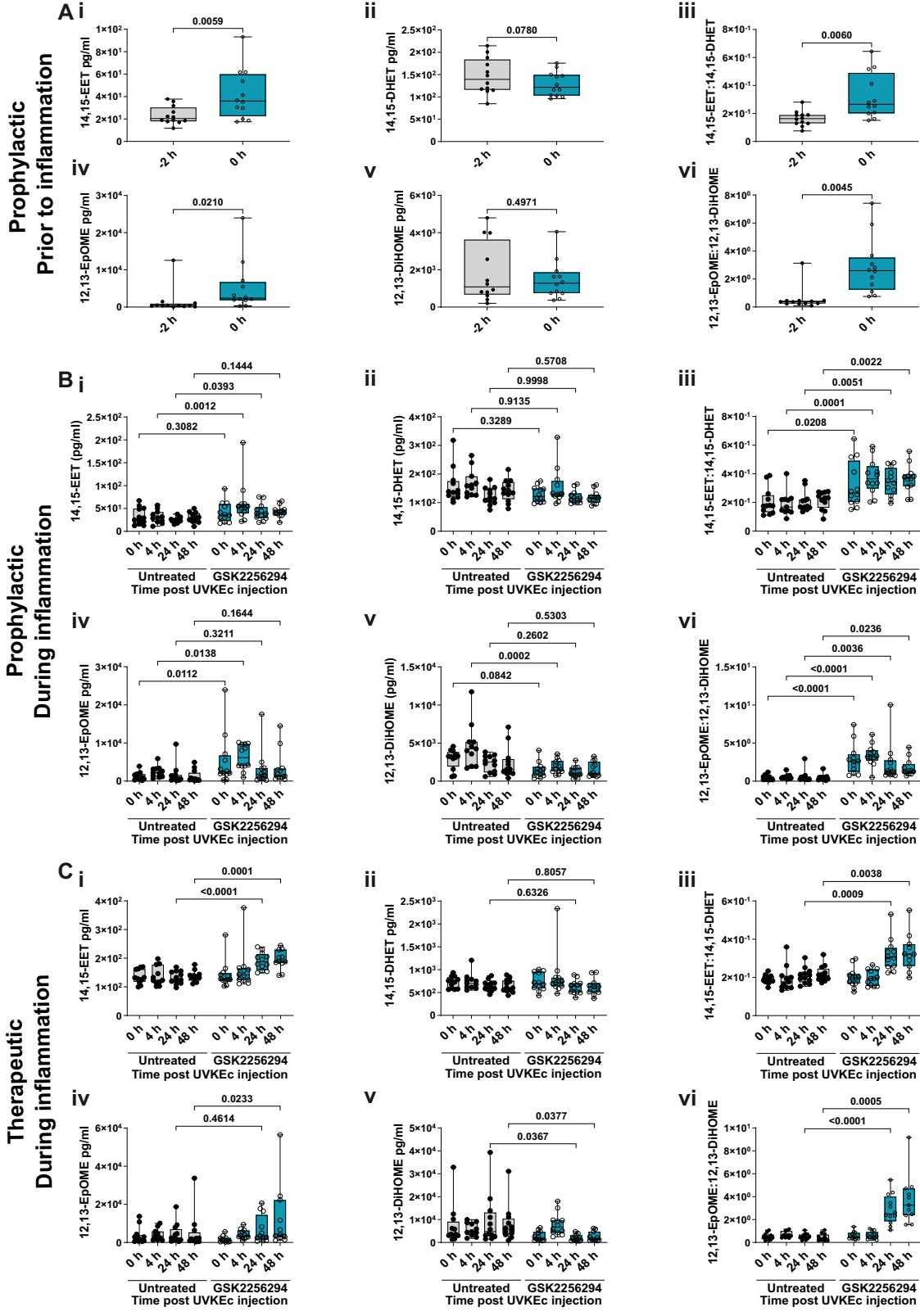

its target p38 MAPK controls the differentiation of classical to intermediate monocytes in humans.

## Inhibiting sEH reduces the number of IMs at the inflammatory site

As in blood, prophylactic dosing caused a significant increase in 12,13-EpOME compared to untreated controls and a concomitant reduction

in 12,13-DiHOME at 4 h, resulting in a significant increase in the ratio of 12,13-EpOME:12,13-DiHOME, similar to the effect seen in blood, Fig. 6A. Therapeutically, GSK2256294 increased the ratio of 12,13-EpOME:12,13-DiHOME at both 24 and 48 h as a result of the significant reduction of 12,13-DiHOME following sEH inhibition (Fig. 6A).

We applied a similar unbiased analytical approach as in Fig. 4 to identify mononuclear phagocytes at the site of inflammation with the

**Fig. 3 | Cytochrome P450-derived lipids are elevated with prophylactic and therapeutic sEH inhibition in plasma.** The forearms of participants were intradermally injected with UV-killed *E. coli* (UVKEc), resulting in a local and peripheral inflammatory response. Two hours prior to (prophylactic) or 4 h after (therapeutic) UVKEc injection, participants were dosed with GSK2256294. Plasma was collected and subject to lipidomic analysis at all timepoints. **A** Quantification of (i) 14,15-EET, (ii) 14,15-DHET, (iii) the ratio of 14,15-EET:14,15-DHET, (iv) 12,13-EpOME, (v) 12,13-DiHOME and (vi) 12,13-EpOME:12,13-DiHOME prior to inflammation in participants dosed with GSK2256294 for 2 h (untreated: $n = 12$; GSK2256294: $n = 12$; biologically independent samples). **B** Quantification of (i) 14,15-EET, (ii) 14,15-DHET, (iii) the ratio of 14,15-EET:14,15-DHET, (iv) 12,13-EpOME, (v) 12,13-DiHOME and (vi) 12,13-EpOME:12,13-DiHOME during inflammation in the prophylactic arm of the study (untreated: $n = 11$–$12$; GSK2256294: $n = 12$; biologically independent samples). **C** Quantification of (i) 14,15-EET, (ii) 14,15-DHET, (iii) the ratio of 14,15-EET:14,15-DHET, (iv) 12,13-EpOME, (v) 12,13-DiHOME and (vi) 12,13-EpOME:12,13-DiHOME in the therapeutic arm of the study (untreated: $n = 11$–$12$; GSK2256294: $n = 11$–$12$; biologically independent samples). All data are expressed in pg/ml. Box-and-whisker plots show the median (centre line), the interquartile range (25th–75th percentiles; box), and the full data range (whiskers, minimum to maximum). Normality was assessed using the D'Agostino & Pearson test, the Shapiro–Wilk test and visualised using a QQ plot. Data in (**A**) were analysed using a two-tailed, paired, parametric t-test. Data in (**B**, **C**) were analysed using two-way ANOVA mixed effect analysis with Šídák's multiple comparison test. Source data are provided as a Source Data file.

gating strategy shown in Supplementary Fig. 4. Here, we found nine separate populations of mononuclear phagocytes in the tissue. CD163$^{hi}$TIM-4$^{lo}$, CD163$^{hi}$TIM-4$^{hi}$, CD206$^{hi}$, classical monocytes (CM), classical-intermediate monocytes (CM-IM), intermediate monocytes (IM), intermediate to nonclassical (IM-NCM) monocyte transitioning cells, non-classical monocytes (NCM) and finally CD205$^{hi}$ Monocyte-DC (Mo-DC) (Fig. 6B) with expression markers for each in Fig. 6C. Monocyte marker expression changes between peripheral blood and tissue, hence the difference in monocyte populations between Figs. 4B and 6C (Supplementary Fig. 9). Boxplots for marker expression in the tissue can be found in Supplementary Fig. 9. Faceting the UMAP of mononuclear phagocyte populations by time and treatment indicated a reduction in intermediate monocytes at 24 and 48 h following GSK2256294 (Fig. 6D). This was confirmed using traditional gating revealing a trend towards a decrease at 24 h in the prophylactic participants and a significant reduction in intermediate monocytes at 48 h in therapeutic participants, which is mirrored in the supervised clustering analysis (Fig. 6E). Classical and non-classical monocyte numbers were not significantly altered in response to dosing with GSK2256294 in both prophylactic and therapeutic dosing regimens (Supplementary Fig. 10).

Compared to untreated controls, prophylactic and therapeutic GSK2256294 had no effect on typical inflammatory cytokines or chemokines including TNFα, IL-10, IL-1β and IL-8 as well as the chemokines involved in monocyte trafficking namely MCP-1, IP-10 (CXCL10) and fractalkine (CX3CL1) (Supplementary Table 1).

**Intermediate monocytes, a double-edged sword in inflammation**

With a reduction in tissue intermediate monocyte-derived macrophages came the opportunity to investigate the role these cells play during resolving inflammation. Other cells that increase during the inflammatory response are CD4 and CD8 T cells, as previously published[35] with GSK2256294 causing a trend towards a reduction in numbers of CD4 T cells and regulatory T cells, but not CD8 cells (Fig. 7A). This reduction was associated with an increase in numbers of total T cells acquiring a vital stain indicating death (Fig. 7A). This occurred alongside increased levels of IL-1α, which is an alarmin liberated into the extracellular space following cell death[38] (Fig. 7B). With intermediate monocytes being the most abundant mononuclear phagocyte at 24 and 48 h (Fig. 7C) co-existing with infiltrating CD4 and CD8 T cells (Fig. 7D), we incubated intermediate monocytes with blood CD4 T lymphocytes finding that intermediate monocytes drive a phenotype reminiscent of activated, proliferating, tissue-resident, T cells with regulatory features (Fig. 7E and Supplementary Fig. 11). Representative flow plots can be found in Supplementary Fig. 12.

In contrast, incubating intermediate monocytes with CD8 T lymphocytes in a CD8 cytotoxicity assay against K562 cells, an MHC-deplete target cell line, caused increased cell death (Fig. 7F). With considerable evidence for expanded intermediate monocytes in diverse chronic inflammatory conditions[39–45] implicated in tissue damage, these data highlight that IMs increase the cytotoxic capacity of CD8 T cells, but also drive CD4 T cell viability and differentiation, on the other.

## Discussion

In this study we characterised the epoxy-oxylipin biosynthetic machinery in humans under baseline and inflammatory conditions demonstrating that blocking sEH significantly elevated 12,13-EpOME and 14,15-EET. With little effect on the salient features of inflammation, except for accelerated pain resolution, sEH inhibition most notably reduced numbers of intermediate monocytes in blood and in inflamed tissue via the inhibition of p38 MAPK by 12,13-EpOME. Reduced intermediate monocytes during tissue resolution uncovered potential a role for these cells in maintaining CD4 T cell viability and phenotype on the one hand, but also revealed their ability to drive cells death via cytotoxic CD8 T cells on the other. With clinical studies demonstrating that sEH inhibition is safe and well tolerated[31,46], therefore, sEH inhibition presents a hitherto unappreciated way of reducing inflammatory intermediate monocytes, which are implicated in the pathogenesis of chronic inflammatory disease.

The major CYP450 isoforms responsible for epoxy-oxylipin synthesis throughout inflammation and resolution are the CYP2C family and CYP2J2[9]. In tissue, these enzymes were found to be elevated with infiltrating immune cells mirroring the biosynthesis of epoxy-oxylipins over time peaking at 14 h, the transition from onset to resolution (Fig. 1A). In parallel, tissue levels of sEH were also elevated, most likely working in tandem to control levels of EpOMEs and EETs (Fig. 1B). Despite the theoretical presence of >20 epoxy-oxylipins and their metabolites in humans, the outcome of this study tells us that during acute dermal inflammation, sEH is primarily responsible for catabolising 12,13-EpOME and 14,15-EET (Fig. 3).

We have shown that during acute inflammation in humans, sEH inhibition reduced numbers of intermediate monocytes in blood and the subsequent appearance of intermediate monocytes in inflamed tissue (Figs. 4 and 6). These findings support a model of sequential monocyte infiltration into inflamed tissues in humans with classical monocytes followed by the intermediate monocytes. As GSK2256294 had no effect on surface expression of intermediate monocyte CCR2, CX3CR1, CXCR3, ICAM-1, VCAM-1 and PSGL-1 (Fig. 4) or levels of their cognate ligands in the blister fluid including CCL2, CXCL10, or CX3CL1 (Supplementary Table 1), we argue that the reduction in inflamed skin intermediate monocytes most likely arose from fewer of these cells appearing in blood following GSK2256294 and not these cells marginating along the luminal aspect of the vasculature.

Contrary to expectation, inhibition of sEH had little effect on pro-inflammatory cytokine levels (Supplementary Table 1). This is in contrast to earlier reports that 14,15-EET, for instance, can block p38 MAPK[15], NLRP3 inflammasome[47] and NFκB activation[16]. It's unclear why blocking sEH had little effect on acute inflammatory soluble mediators and cardinal signs of inflammation, except for pain, only to highlight that our current data are from a complex human model instead of isolated cell systems and rodents.

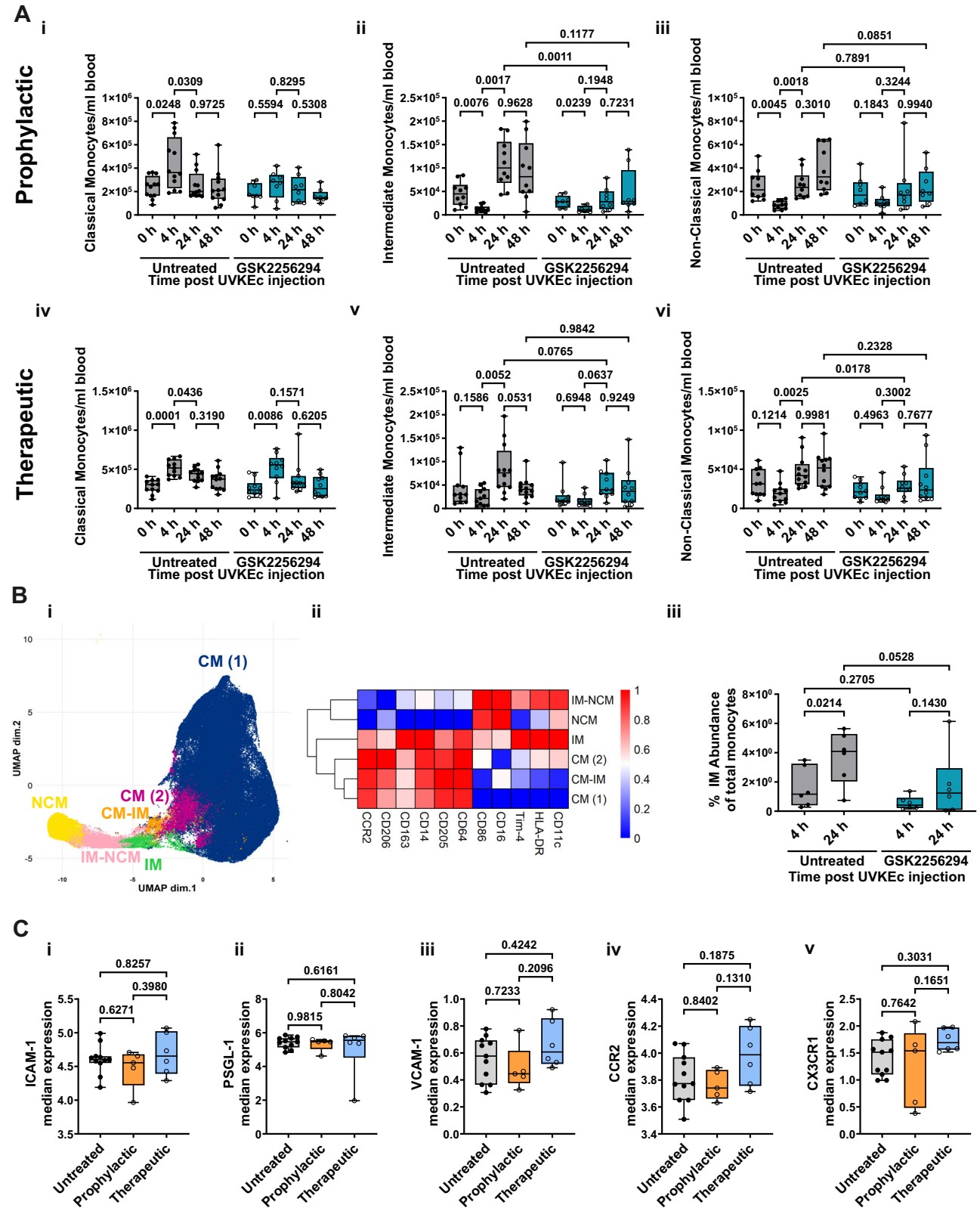

Originally, classical (CD14⁺CD16⁻) and non-classical (CD14⁻CD16⁺) monocytes were identified[48] followed by a population that is intermediate and expresses both CD14 and CD16 whilst also bearing a distinct transcriptomic profile[49,50]. These 'intermediate type' monocytes displayed comparable ROS production and phagocytosis potential, but higher class II molecule expression and IL-12 production than classical monocytes as well as secretion of TNF-α, IL-1β, IL-6 and CCL3

upon TLR stimulation[50,51]. Mathematical modelling of monocyte differentiation, as well as kinetic and deuterium labelling studies, demonstrated a linear trajectory from classical to non-classical monocytes[33,36,37]. However, the molecular mechanisms controlling monocyte differentiation are poorly understood. Previous studies have found that epoxy-oxylipins reduce the phosphorylation of p38 MAPK[52–55]. p38 MAPK is hyperphosphorylated in CM, with its

**Fig. 4 | sEH inhibition both prophylactically and therapeutically inhibits the expansion of intermediate monocytes.** The forearms of participants were intra-dermally injected with UV-killed *E. coli* (UVKEc), resulting in a local and peripheral inflammatory response. Two hours prior to (prophylactic) or 4 h after (therapeutic) UVKEc injection, participants were dosed with 15 mg of GSK2256294. Peripheral blood was collected, and leucocytes analysed at baseline and 4, 24 and 48 h post UVKEc injection using multiparameter flow cytometry. **A** Classical (CD14+CD16−), intermediate (CD14+CD16+) and non-classical monocytes (CD14−CD16+) were quantified using manual gating in FlowJo. (i) Classical, (ii) intermediate and (iii) non-classical monocyte numbers during inflammation with prophylactic sEH inhibition (untreated: *n* = 10-12; GSK2256294: *n* = 8; biologically independent samples). (iv) Classical, (v) intermediate and (vi) non-classical monocytes during inflammation with therapeutic sEH inhibition (untreated: *n* = 11–12; GSK2256294: *n* = 10–11; biologically independent samples). **B** UMAP of monocyte populations identified in peripheral blood in the therapeutic arm of the study. Total monocytes were extracted from FCS files in FlowJo and clustered using CATALYST; (i) six monocyte populations were identified; (ii) heatmap of marker expression in each cluster; (iii) the percentage of intermediate monocytes using supervised clustering was quantified as a % of total monocytes at 4 and 24 h in untreated and drug-treated participants. (untreated: *n* = 6; GSK2256294: *n* = 6; biologically independent samples). **C** Intermediate monocytes at 24 h were analysed using CATALYST for median expression of (i) ICAM-1, (ii) PSGL-1, (iii) VCAM-1, (iv) CCR2 and (v) CX3CR1 in untreated, prophylactic and therapeutic participants (untreated: *n* = 11; GSK2256294 prophylactic: *n* = 5; GSK2256294 therapeutic: *n* = 6; biologically independent samples). Data were assessed for normalisation using the D'Agostino & Pearson test, the Shapiro−Wilk test and visualised using a QQ plot. Box-and-whisker plots show the median (centre line), the interquartile range (25th−75th percentiles; box), and the full data range (whiskers, minimum to maximum). Data in (**A**) were analysed using two-way ANOVA mixed effect analysis with Tukey's multiple comparisons test. Data in (**B**) were analysed using two-way ANOVA mixed effect analysis with Uncorrected Fisher's LSD. Data in (**C**) were analysed using a One-Way ANOVA. Source data are provided as a Source Data file.

expression decreasing as these cells differentiate (Fig. 5B). We have shown that 12,13-EpOME reduces phosphorylation of p38 MAPK, which in turn blocks monocyte differentiation, albeit in vitro (Fig. 5A). Furthermore, the p38 MAPK inhibitor, losmapimod, caused a significant reduction in intermediate monocytes in human volunteers after oral dosing, collectively showing that p38 MAPK, at least in part, drives CM-IM differentiation and which is counter-regulated by 12,13-EpOME (Fig. 5E).

Our findings allow us to understand what role intermediate monocytes might have during the resolution of acute inflammation. Numbers of CD4 T cells were reduced at 48 h following GSK2256294, coincident with an increase in T cells acquiring a live dead stain and remnants of immune debris. Given that IMs are significantly higher in number compared to CM (Fig. 7C)−350 CM versus 1800 IMs−and that only IMs are reduced in the skin following GSK2256294, coincident with reduced T cells (Fig. 7A), we logically concluded that during resolving inflammation, when IM predominate, they maintain CD4 T cell viability (Fig. 7E).

By corollary, we found that prolonged culture of intermediate monocytes with CD8 T cells caused marked cytotoxic activity against an MHC-deplete cell line. These studies were prompted by reports of unrestrained expansion of CD16+ monocytes being a hallmark of many diseases driven by chronic inflammation. For instance, there is a relationship between intermediate monocyte numbers and elevated predictors for cardiovascular disease risk[41]. Obesity has also been shown to induce monocytosis of the intermediate and non-classical subsets, while transcriptomic analysis of monocytes in obese donors demonstrates increased expression of TLR4 and TLR8 and secretion of pro-inflammatory cytokines such as IL-1β and TNFα in response to LPS or ssRNA stimulation[44]. Moreover, intermediate monocytes are enriched in rheumatoid arthritis patients (causing Th17 cell expansion[43]), lupus[45], Graves' disease[40], HIV[39] and tuberculosis[42]. Thus, we wished to contextualise the properties of intermediate monocytes depending on when they transiently appear during resolving inflammation maintaining CD4 T cell viability, to when they persist at sites of chronic inflammation where they turn pathogenic and contribute to tissue damage. With the World Health Organisation ranking diseases driven by chronic inflammation as the greatest threat to human health, our serendipitous finding that sEH controls the expansion of cells that are purported to be pathogenic in many chronic inflammatory diseases, suggested that inhibition of sEH may represent a tractable approach to treating chronic inflammation.

In summary, this is the first study to chart the expression of cytochrome P450/EH enzymes and their epoxy-oxylipin biosynthesis during acute inflammation in humans, finding new biology surrounding monocyte differentiation whilst putting into perspective the role of IM population in health and disease.

## Methods

### Ethical statement
All studies were approved by the UCL Research Ethics Committee (Project ID number: 1309/004). All procedures performed were in line with the ethical standards of the UCL Institutional Committee and according to the principles of the Declaration of Helsinki. Written informed consent was obtained from all volunteers prior to their participation.

### Volunteer recruitment
In line with our ethical approval, we recruited young, healthy, male volunteers who were non-smokers between the ages of 18−50 for this study. Sex of participants was determined based on self-reporting. Exclusion criteria included vaccination in the previous three months, allergies, diagnosis of chronic inflammatory disease, taking regular prescribed medication, or taking over-the-counter medication within 7 days prior to their participation in the study. Prior to the study commencing all volunteers were given a participant information sheet and required to attend a screening session to assess eligibility where written consent was obtained. At the screening session, bloods were taken and sent to The Doctor's Laboratory (TDL, London, UK) to record full blood count and biochemistry. For the 48-h duration of the study, participants were asked to abstain from alcohol and caffeine to control for the effect these drugs have on inflammatory processes. Participants were compensated for their time in accordance with ethical approval. No statistical methods were used to predetermine sample sizes. As this experimental study was designed to test a hypothesis in humans in vivo, and not to determine clinical benefit, no randomisation was performed. No adverse events were observed in our study group.

### The human intradermal UV-killed *E. coli* inflammatory model
The human intradermal UV-KEc model of inflammation was developed in the host laboratory to study resolving inflammation in response to a localised bacterial insult[32–35]. Briefly, $1.5 \times 10^7$ UV-KEc (Strain: NCTC 10418, Public Health England, UK) in 100 μL saline was administered intradermally into the forearms of participants. This results in localised inflammation characterised by pain, heat, redness, and swelling. Using negative pressure blister suction machines (Electronic Diversities Ltd.), two blisters (one on each forearm) were raised over the inflammatory site to acquire inflammatory cells and exudate for the determination of local immune cells profiles as well as cytokine, chemokine and lipid mediator levels. Blister exudate is collected into 50 μL sodium citrate (3% in PBS). The UV-KEc model also results in systemic changes in peripheral blood, which is collected at each time point for plasma analysis and cellular characterisation.

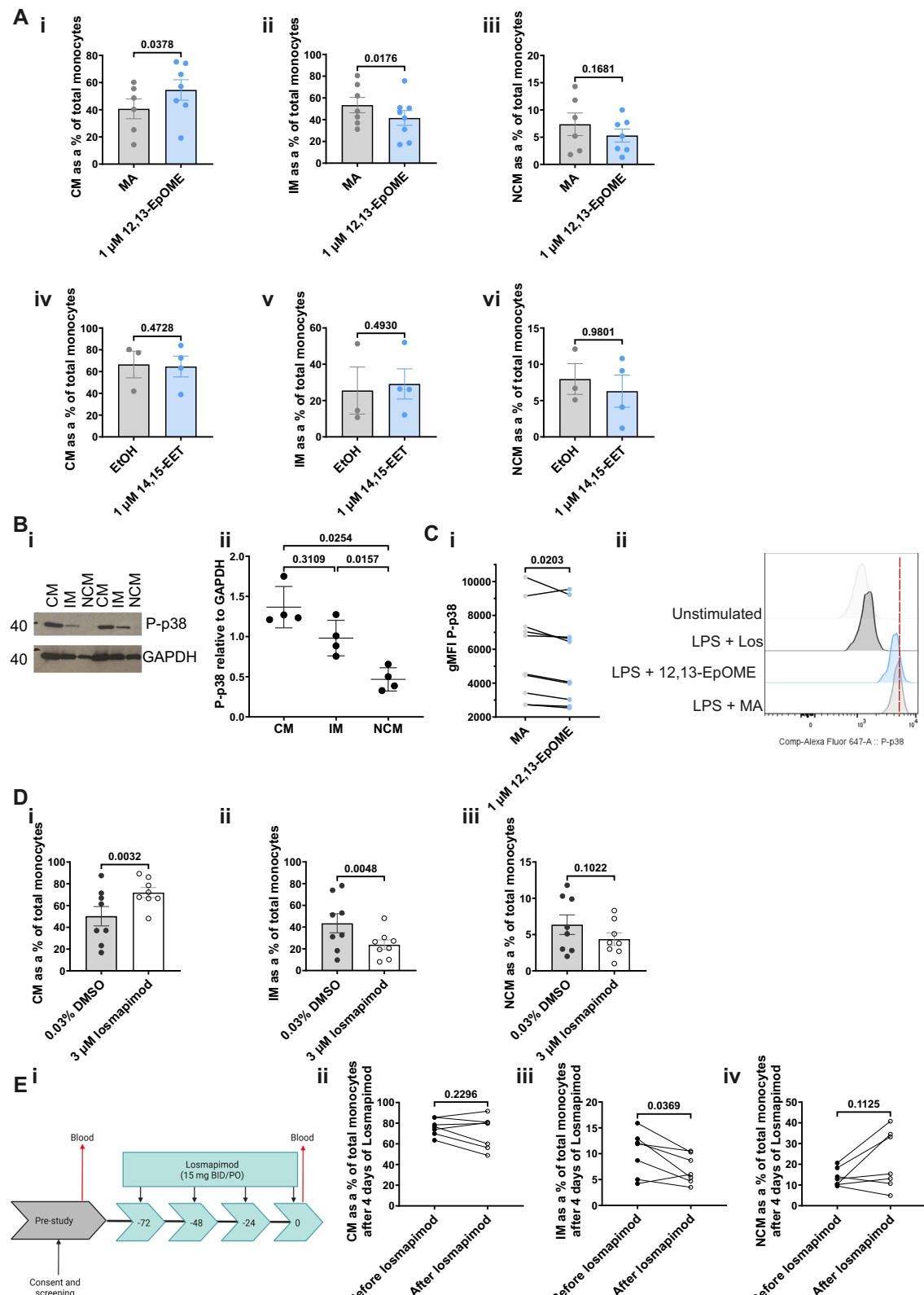

## sEH inhibitor study design

**Prophylactic.** The initial visit (−24 h) for individuals allocated to the sEH inhibited group included pre-dose recording of baseline inflammatory symptoms and blood sample collection. Prior (−2 h) to intradermal injection of UV-killed *E. coli*, participants were orally dosed with 15 mg of the sEH inhibitor GSK2256294. Untreated participants did not receive a placebo control. At the zero hour time point, dermal inflammation was induced. The protocol for the subsequent visits for the sEH-inhibited volunteers was identical to the control group; at the beginning of every visit (0, 4, 24 and 48 h post-induction of inflammation) inflammatory symptoms were evaluated (discussed below), and peripheral blood was collected at each timepoint for plasma, cellular analysis by flow cytometry, and storing PBMCs. At the 4 and 24 h time points, the inflammatory site was sampled by the creation of suction blisters.

**Fig. 5 | The effect of 12,13-EpOME and p38 inhibition on monocyte differentiation. A** PBMCs ($2 \times 10^6$/well) were seeded and treated with 1 μM epoxy-oxylipin or vehicle for 24 h. Cells were phenotyped by flow cytometry. (i) % classical monocytes (CM), (ii) % intermediate monocytes (IM), (iii) % non-classical monocytes (NCM) with 1 μM 12,13-EpOME vs. methyl acetate (MA) (MA: $n = 6$; 12,13-EpOME: $n = 7$; biologically independent samples). (iv–vi) % CM, IM, and NCM with 1 μM 14,15-EET vs ethanol (EtOH) (EtOH: $n = 3$; 14,15-EET: $n = 4$; biologically independent samples). **B** (i) Representative image of a western blot for P-p38 in monocyte subsets. (ii) Quantification of P-p38 expression in monocyte subsets relative to GAPDH ($n = 4$; biologically independent samples). **C** PBMCs ($2 \times 10^6$/well) were treated with 1 μM 12,13-EpOME or vehicle (MA) for 24 h, then re-stimulated with 1 μM 12,13-EpOME or MA plus 100 ng/ml LPS for 30 min. (i) Phosphorylation of p38 in PBMCs treated with 1 μM 12,13-EpOME or MA ($n = 10$; biologically independent samples). (ii) Representative histogram of P-p38 MFI in PBMCs treated with LPS + MA, LPS + 12,13-EpOME, LPS + losmapimod (Los), or unstimulated. **D** PBMCs ($2 \times 10^6$/well) were treated with 3 μM losmapimod or DMSO (vehicle) for 24 h, then phenotyped by flow cytometry. Shown are % (i) classical (CM), (ii) intermediate (IM), and (iii) non-classical monocytes (NCM) ($n = 8$; biologically independent samples). **E** Older people (>65 years old) were dosed with losmapimod daily for four days. Peripheral blood was taken both before and after dosing with losmapimod and analysed for monocyte subsets using flow cytometry. % of (i) CM, (ii) IM, (iii) NCM in peripheral blood at steady state before and after dosing with losmapimod ($n = 8$; biologically independent samples). Data were assessed for normalisation using the D'Agostino & Pearson test, the Shapiro–Wilk test and visualised using a QQ plot. Parametric data are presented as mean ± SD. Data in (**A**, **C**, **D**, **E**) were analysed using a two-tailed parametric, paired t-test. Data in (**B**) were analysed using a one-way ANOVA. Source data are provided as a Source Data file.

**Therapeutic.** Participants attended sessions in the lab four times over the course of the experiment: at 0 h to induce inflammation, 4, 24 and 48 h after the onset of inflammation. Participants in the drug arm of the study received a single 15 mg dose of GSK2256294 at the 4 h timepoint after the UV-KEc injection. Untreated participants did not receive a placebo control. The study was divided into two flow cytometry panels. Panel 1 was for the investigation of monocytes and dendritic cells at the site of inflammation and panel 2 was for investigating T-cell subsets. Studies using panel 1 involved participants receiving one blister at 24 h and the second blister at 48 h. Studies using Panel 2 involved participants receiving two blisters at 48 h, one on each forearm. Peripheral blood was collected at each time point for plasma, cellular analysis by flow cytometry, and storing PBMCs.

### Non-invasive inflammatory symptom recordings
Core temperature, ambient temperature, and temperature at the injection site were monitored at each time point. Participants were asked for a pain score at the inflammatory site between 0 and 10. Additionally, participants were asked for a tenderness score. For this, a 100 g weight was placed onto the forearm at the centre of the inflammatory site, and participants were asked to give a tenderness score of between 0 and 10. Finally, changes to vascular hyperaemia were investigated through the use of a Laser Doppler Imager (Moor LDI-HIR, Moor instrument). The laser emitted by the scanner is scattered by red blood cells present at the site of inflammation. The change in the frequency of light in the laser beam is detected by a photodetector. The signal is measured in arbitrary perfusion units; the 'intensity' of which is dependent on the velocity and concentration of the red blood cells in the fixed area being scanned. Images were analysed using the LDI software v5.2. To exclude background noise, the threshold of the perfusion unit was set to >300.

### Flow cytometry
To isolate leucocytes for flow cytometry analysis, 1 ml of whole blood, collected in a EDTA vacutainer tube (4 ml; Griener Bio-One), was added to 9 ml of ammonium-chloride-potassium lysis buffer (ACK; Lonza). Blister cells were obtained via centrifugation of blister exudate. All samples were stained in 100 μL for 30 min at 4 °C. Antibody cocktails were prepared with 50 μL of Brilliant Stain Buffer (BD Biosciences, 566349), antibodies (Supplementary Table 2), with the remainder of the volume made up with FACS buffer (PBS, 5% Foetal Calf Serum, 2 mM EDTA). All panels were compensated appropriately, either using cells as vehicles for single stains, UltraComp eBeads™ compensation beads (Thermo Fisher, 01-2222-41), or ArC™ Amine Reactive compensation bead kit (Thermo Fisher, A10346). Unstained sample and Fluorescence minus-one (FMO) and, where appropriate, isotype control antibodies were used as a control. Data were acquired on a BD Fortessa UV X20 flow cytometer using BD FACS DIVA software or a Sony ID7000 spectral flow cytometer. Data analysis was performed using FlowJo software v10.10 or using the R package 'CATALYST'

(v1.26.1)[56]. Blister cell numbers were quantified by acquiring the entire sample on the flow cytometer. Peripheral blood samples were quantified either by manual haemocytometer or CountBright cell counting beads (Invitrogen).

### Isolation of peripheral blood mononuclear cells (PBMCs)
Peripheral blood was collected in EDTA. Peripheral blood mononuclear cells were isolated by mixing 1:1 with Hanks Buffer Salt Solution (HBSS) (Gibco, 14170). The 1:1 peripheral blood/HBSS solution was layered onto Ficoll-Paque PLUS (17-1440-03) in a 2:1 ratio. Samples were spun at $1000 \times g$ for 30 min with the brakes off. The layer of PBMCs was collected and washed in HBSS, followed by a spin at $400 \times g$ for 10 min. Red blood cell contamination was lysed using 1 ml Ack lysis buffer for 5 min at RT. Samples were topped up to 30 ml with PBS, cells were enumerated and then spun for a further 5 min at $400 \times g$.

### In vitro monocyte differentiation assay
Following isolation of PBMCs, samples were plated or immediately stained for analysis by flow cytometry. Samples were plated in RPMI complete medium (RPMI (31870-025), 10% AB human serum (H4522), 1% pen/strep (15140-122), 1% L-glutamine (15140-122)) at a cell density of 2 million cells/well in a 6 well untreated plate (Corning, 3736) or 1 million cells/well in a 24 well untreated plate (Corning, 3738) for 24 h at 37 °C incubator either alone or with different treatments. To test the effect of epoxy-oxylipins on monocyte conversion, 12,13-EpOME (Cayman Chemical, 52450), 12,13-DiHOME (Cayman Chemical, 10009832), 14,15-EET (Cayman Chemical, 50651) and 14,15-DHET (Cayman Chemical, 51651) were added to PBMCs at a concentration of 1 μM. Lipids had their respective vehicles (Methyl Acetate (MA) for EpOMEs/DiHOMEs and ethanol (EtOH) for EETs/DHETs) evaporated under nitrogen gas in glass vials and were immediately reconstituted in RMPI complete medium prior to addition to the plate. Vehicle controls were prepared by adding the respective amount of MA or EtOH into a glass vial, which was then evaporated under nitrogen gas, followed by the addition of RMPI complete medium to control for any residual vehicle. Losmapimod was reconstituted in DMSO and diluted in RMPI complete medium and added to wells at a final concentration of 3 μM. A 0.03% DMSO vehicle was used. After a 24 h incubation period PBMCs were lifted from the plate and pelleted at $400 \times g$ for 5 min. Dissociation buffer (EDTA + Lidnocaine) was added to the wells and incubated for 5 min at RT. Pellets were resuspended in dissociation buffer (containing residual cells) and spun for a further 5 min at 400 g. Monocyte subsets were stained and visualised by flow cytometry as CD3−CD19−CD20−CD56-HLA-DR+CD14+CD16+.

### Phosphorylation assays
To examine the phosphorylation status of proteins in monocytes, PBMCs were isolated and seeded into 24-well plates at a density of $1 \times 10^6$ cells per well. PBMCs were cultured with 1 μM 12,13-EpOME, MA

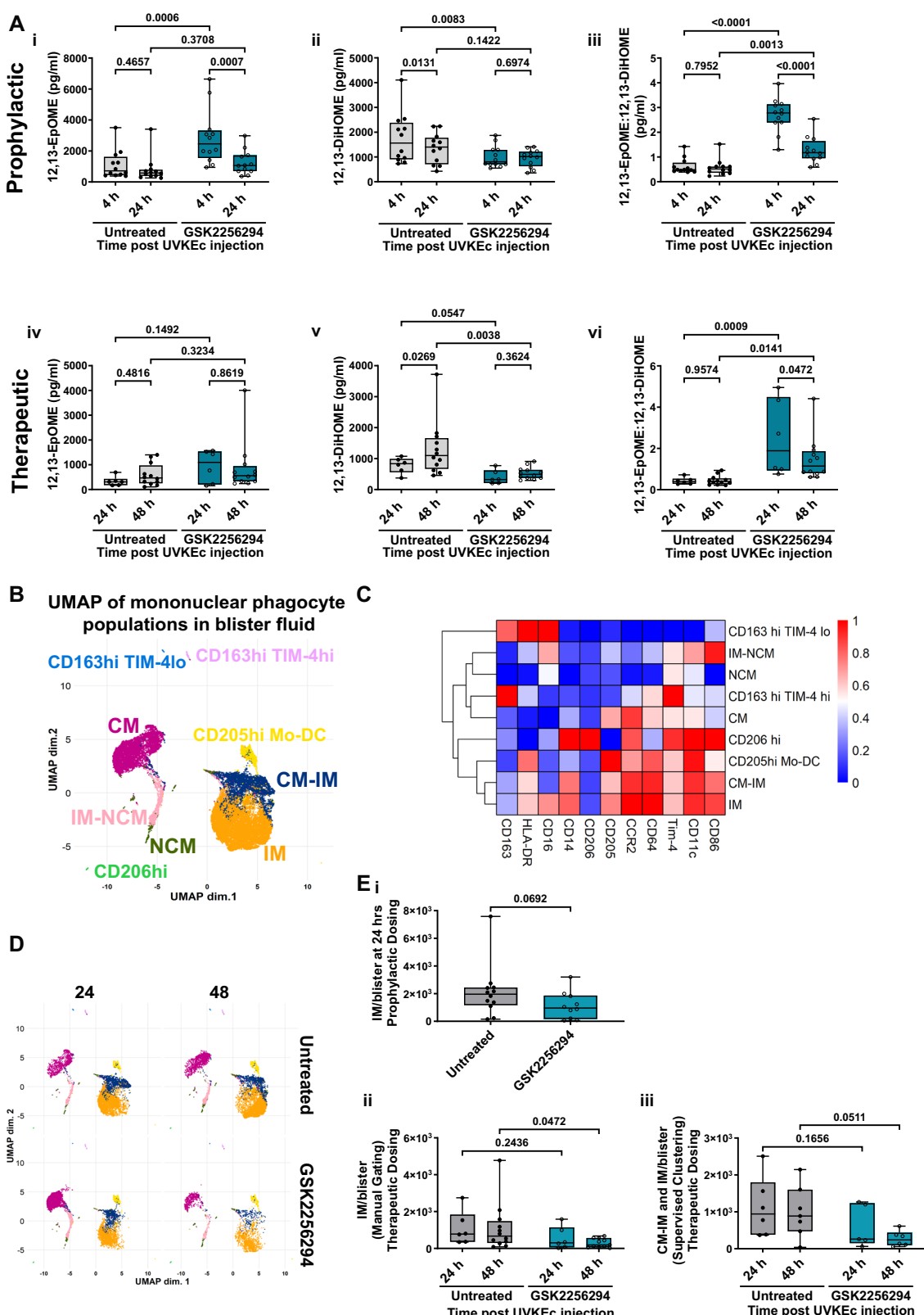

or losmapimod overnight. After a 24 h incubation period, lipids or losmapimod were added again and incubated with 100 ng/LPS for 30 min. After 30 min, 4% PFA was added directly to wells at a 1:1 ratio and incubated for 15 min at 37 °C to preserve phosphorylation status. Cells were spun at 400 g for 5 min, the supernatant removed and then washed in 300 µL perm buffer and spun again. Samples were resuspended in 200 µL fixation buffer (Thermofisher, 00-5523-00) and

incubated for 10 min at RT in the dark. Samples were spun at $400 \times g$ for 5 min and subsequently washed in perm buffer and then stained for AF647 phospho-p38 (1:10, 526066).

**Immunohistochemistry**
To analyse the temporal profile of both the CYP and EH enzymes, skin biopsies were obtained for immunohistochemistry analysis.

**Fig. 6 | sEH inhibition significantly increases the ratio of 12,13-EpOME:12,13-DiHOME and reduces numbers of intermediate monocytes at the inflammatory site.** The forearms of participants were intradermally injected with UV-killed *E. coli* (UVKEc) resulting in a local inflammatory response. Two hours prior to (prophylactic) or 4 h after (therapeutic) UVKEc injection, participants were dosed with 15 mg of GSK2256294. Local inflammatory exudate was collected and subject to lipidomic and flow cytometric analysis at 4 and 24 h (prophylactic) and 24 and 48 h (therapeutic). **A** Quantification of (i) 12,13-EpOME, (ii) 12,13-DiHOME and (iii) 12,13-EpOME:12,13-DiHOME in blister fluid at 4 and 24 h (prophylactic) (untreated: $n = 12$; GSK2256294: $n = 12$; biologically independent samples). Quantification of (iv) 12,13-EpOME, (v) 12,13-DiHOME and (vi) 12,13-EpOME:12,13-DiHOME in blister fluid at 24 and 48 h (therapeutic) (untreated: $n = 6$; GSK2256294: $n = 12$; biologically independent samples). Data are expressed in pg/ml. **B** UMAP of monocyte populations identified in local inflammatory exudate (therapeutic arm). Total monocytes were extracted from FCS files in FlowJo and analysed using CATALYST. Eight monocyte populations were identified and labelled. **C** Heatmap of marker expression in each cluster. **D** UMAP of monocyte populations faceted by time and treatment (**B**–**D** representative of untreated: $n = 6$; GSK2256294: $n = 6$–7; biologically independent samples). **E** Classical monocytes (CD14+CD16−), intermediate monocytes (CD14+CD16+) and non-classical monocytes (CD14−CD16+) were quantified using manual gating in FlowJo. (i) Intermediate monocytes/blister at 24 h (prophylactic) (untreated: $n = 12$; GSK2256294: $n = 10$; biologically independent samples). (ii) Intermediate monocytes/blister at 24 and 48 h (therapeutic) (untreated: $n = 6$–12; GSK2256294: $n = 6$–11; biologically independent samples). (iii) CM-IM and IM populations from the supervised clustering analysis were combined and analysed as a % of total monocytes at 24 and 48 h (therapeutic) (untreated: $n = 6$; GSK2256294: $n = 6$–7; biologically independent samples). Normality was assessed using the D'Agostino & Pearson test, the Shapiro–Wilk test and visualised using a QQ plot. Box-and-whisker plots show the median (centre line), the interquartile range (25th–75th percentiles; box), and the full data range (whiskers, minimum to maximum). Data in **A**, **E**(ii–iii) were analysed using two-way ANOVA mixed effect analysis with Uncorrected Fisher's LSD. Data in **E**(i) were analysed using a two-tailed, Mann–Whitney test. Source data are provided as a Source Data file.

Acu-Punch 3-millimetre kits (Schuco) were used post anaesthetisation (with 1% lignocaine hydrochloride (Hameln)) to collect skin biopsies at baseline or post UV-KEc injection. The biopsies were washed with saline (Hameln) and fixed for 24 h in 10% neutral buffered formalin (Roche) at 4 °C. Samples were embedded in paraffin, and 4 μm-thick sections were cut and placed on glass slides. Antibodies used for immunohistochemistry can be found in Supplementary Table 3.

Initially, the majority of paraffin was removed from paraffin-embedded biopsy sections by melting at 75 °C for 15 min. The remainder was removed by using an Autostainer (Leica); the biopsy sections were washed in xylene (a solvent) for 20 min. To remove the xylene, sections were washed with 100% ethanol for 10 min. To hydrate the sections, 5-min graded washes of ethanol (100, 70 and 30% ethanol) to water were performed. Antigen retrieval was achieved by breaking the formalin cross-links with a sodium citrate solution (0.3 % sodium citrate (Sigma-Aldrich), 0.05 % TWEEN 20 (Sigma-Aldrich), pH 6) at 95 °C in the water bath for 13 min. Following antigen retrieval, sections were washed with PBS and permeabilised/blocked for endogenous peroxidase activity by incubation with 3% hydrogen peroxide (VWR Chemicals) in methanol (VWR Chemicals) for 10 min at room temperature. The primary polyclonal antibody (supplementary) dilutions were prepared with antibody diluent PBS + 0.01% sodium azide (Sigma-Aldrich) + 0.1% bovine serum albumin (Sigma-Aldrich) + 4% horse serum (Thermo Fischer Scientific) and the sections were incubated with the primary antibody overnight at 4 °C. Subsequently, the sections were washed with PBS and incubated with a horseradish peroxidase anti-rabbit immunoglobulin G (IgG) polymer reagent biotinylated secondary antibody; ImmPRESS™ horseradish peroxidase Anti-Rabbit IgG (Polymer Detection Kit) for 30 min at room temperature. To visualise staining, the ImmPACT® NovaRED™ Peroxidase Substrate kit was used. Finally, sections were incubated in haematoxylin for 6 s for counterstaining. Excess haematoxylin staining was removed by incubation with acidified (hydrochloric acid; Thermo Fischer Scientific) 70% ethanol for 10 s. The sections were washed with water for 2 min and dehydrated by incubation with 70% and 100% ethanol for 2 and 4 min, respectively. To protect the biopsies, cover slips were applied using xylene and an automated Coverslipper machine (Leica). To analyse, sections were scanned using the Nanozoomer (Hamamatsu). The magnified images were taken using a 100x lens on a brightfield microscope (Euromex Novex 86025-LED). The Leica LAS AF Lite software v4.0, NDP.view v2.2 and imageJ v2 were used to qualitatively analyse images.

## Lipidomics
Peripheral blood was collected by venepuncture using an aseptic technique. Plasma was isolated by centrifugation of sodium heparin vacutainer tubes (10 ml; Griener Bio-One) at $20,000 \times g$ for 10 min (room temperature). Both plasma and blister fluid tubes were frozen at −80 °C, anonymized and sent for lipid analysis at the Zeldin laboratory (National Institute for Environmental Health Sciences). A total of 297 biologically independent samples were analysed. Plasma samples (200 μL) were thawed and diluted 1:1 with (5 % methanol and 0.1 % acetic acid). To prevent epoxy-oxylipin catabolism by sEH during extraction, 1 μL of t-AUCB in methanol (sEH-I; 10 μM final concentration) was added to each sample. After the addition of internal standards (11,12-DHET-d11 (150 ng), 11,12-EET-d11 (300 ng), 15-HETE-d8 (300 ng)), extraction was performed by liquid:liquid extraction using 1 ml ethyl acetate. The samples were dried and reconstituted in 50 μL of 30% glycerol. 10 μL injections were assayed on an Ultimate 3000 UHPLC equipped with an Xselect CSH C18, 2.1 × 50 mm, 3.5 μm particle column (Waters, Wilford, MA) and a TSQ Quantiva tandem mass spectrometer (Thermo Fisher Scientific, Waltham, MA), as previously described[57]. Relative response ratios of analytes were compared to linear standard curves generated with purchased oxylipins (Cayman Chemical, Ann Arbor, MI). Analyte retention times, ionisations, internal standards used and the range and r2 of the standard curves are shown in Supplementary Table 4. Representative chromatograms for 14,15-EET, 14,15-DHET, 12,13-EpOME and 12,13-DiHOME from a plasma sample are presented in Supplementary Fig. 13.

## Luminex cytokine/chemokine analysis
Cytokine and chemokine profiles in blister fluid were quantified using the Luminex Performance Human XL Cytokine Panel (FCSTM18). Blister fluid was applied neat to the plate.

## Losmapimod study design
The losmapimod experimental design has been previously described[58]. Peripheral blood samples were collected, and monocyte subsets were visualised using flow cytometry.

## CD4 T cell−monocyte co-culture
Intermediate monocytes and total CD4 T cells were isolated using a BD S8 Spectral cell sorter (BD Biosciences) with the gating strategy shown in Supplementary Fig. 14, using a panel including Zombie NIR (Biolegend), BUV615 HLA-DR (BD Biosciences), BUV395 CD4 (BD Biosciences), AF700 CD14 (Biolegend), BV650 CD16 (BD Biosciences), APC-H7 CD3 (BD Biosciences). CD4 T cells were cultured with each monocyte subset at a 5:1 ratio for 48 h in complete RMPI medium incubated with cytokines 5 ng/ml IL-6 (Thermofisher, A42541), 20 ng/ml IL-8 (Thermofisher, PHC0084), 10 ng/ml IL-1β (Thermofisher, A42509), 20 ng/ml IL-18 (R&D, 9124-IL-050/CF) and 10 ng/ml IL-15 (Thermofisher, 200-15). Following incubation, samples were lifted from the plate, and the phenotype of CD4 T cells was analysed using a Sony

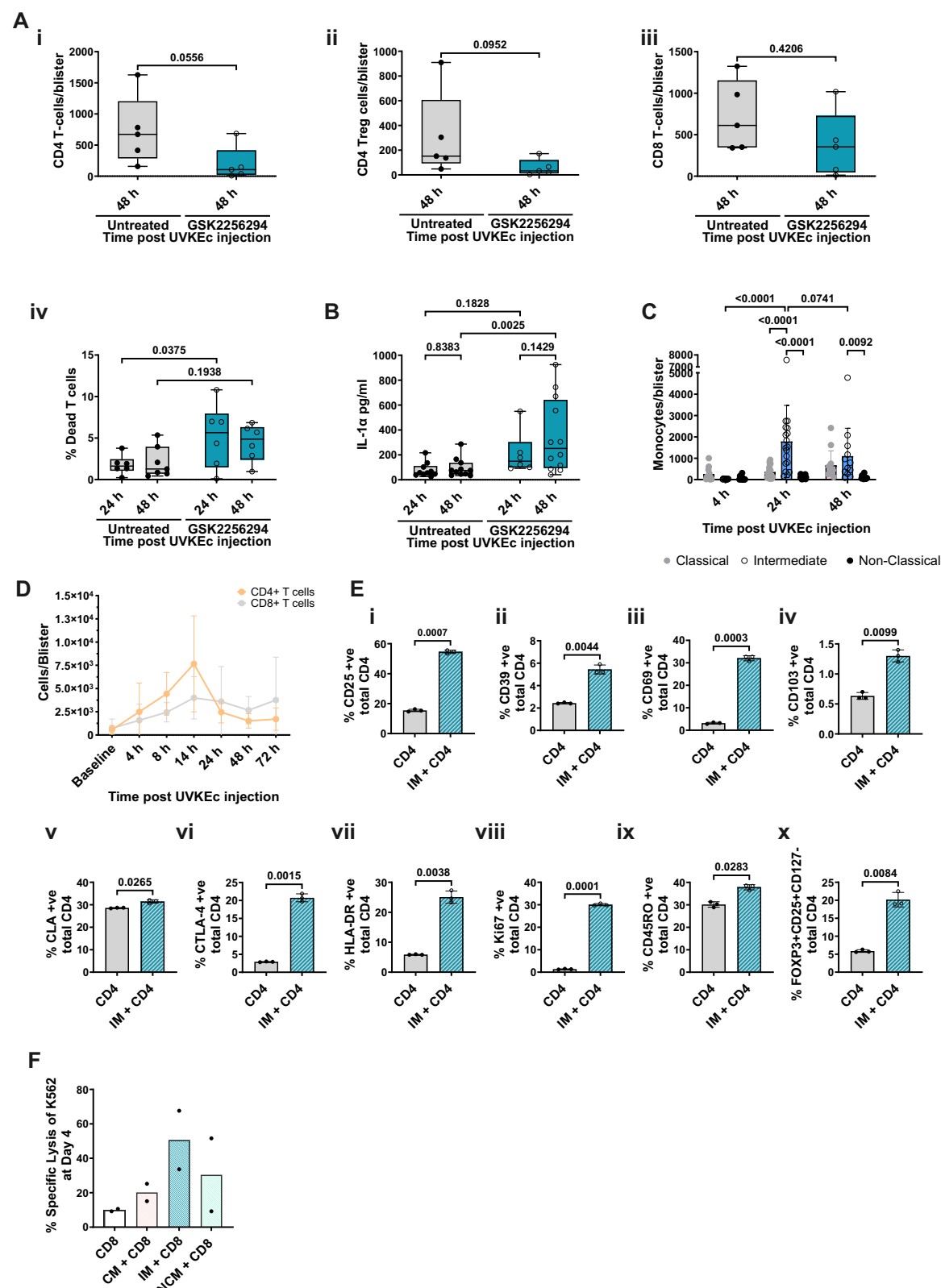

ID7000 spectral flow cytometer with a panel containing antibodies for CD25, CD39, CD69, CD103, CLA, CTLA-4, HLA-DR, Ki67, CD45RO, FOXP3 and CD127 (Supplementary Table 2).

## CD8 cytotoxicity assays
NK, CD8 T cells and monocyte subsets were isolated using a BD S8 Spectral cell sorter (BD Biosciences) using a panel including CD14

BUV805 (BD Biosciences), CD16 BV711 (BD Biosciences), HLA-DR PE (Biolegend), CD56 BUV395 (BD Biosciences), TCRαβ BUV615 (BD Biosciences), CD8 APC-Cy7 (Biolegend), CD19/CD20 FITC (Biolegend) and Live/Dead Zombie UV (Biolegend), plated and incubated with and without cytokines 5 ng/ml IL-6 (Thermofisher, A42541), 20 ng/ml IL-8 (Thermofisher, PHC0084), 10 ng/ml IL-1β (Thermofisher, A42509), 20 ng/ml IL-18 (R&D, 9124-IL-050/CF) and 10 ng/ml IL-15

**Fig. 7 | CD4 T cells are reduced at the inflammatory site with therapeutic sEH inhibition.** The forearms of participants were intradermally injected with UV-killed *E. coli* (UV-KEc). Two hours prior to (prophylactic) or 4 h after (therapeutic) UV-KEc injection, participants were dosed with 15 mg of GSK2256294. Local inflammatory exudate was subject to flow cytometric analysis at 4 and 24 h (prophylactic) and 24 and 48 h (therapeutic). **A** (i) CD4, (ii) CD4 T-regulatory, (iii) CD8 T cell numbers per blister at 48 h (therapeutic) (untreated: $n = 5$; GSK2256294: $n = 5$; biologically independent samples). (iv) % dead T cells at 24 and 48 h (therapeutic) (untreated: $n = 6$-7; GSK2256294: $n = 6$; biologically independent samples). **B** Concentration of IL-1α in pg/ml in blister fluid at 24 and 48 h (therapeutic) (untreated: $n = 10$–11; GSK2256294: $n = 6$–12; biologically independent samples). **C** Monocyte subset numbers/blister at 4, 24 and 48 h in untreated participants ($n = 11$–18; biologically independent samples). **D** Time course of CD4 and CD8 subset numbers/blister in untreated participants ($n = 6$; biologically independent samples). **E** Intermediate monocytes were co-cultured with CD4 T cells at a 5:1 (T cell:monocyte) ratio for 48 h and analysed by spectral flow cytometry. The % expression on total CD4 cells

was visualised for (i) CD25, (ii) CD39, (iii) CD69, (iv) CD103, (v) CLA, (vi) CTLA-4, (vii) HLA-DR, (viii) Ki67, (ix) CD45RO and (x) FOXP3+CD25+CD127− ($n = 3$; technical repeat). **F** Classical, intermediate and non-classical monocytes and CD8 T cells were co-cultured for 4 days at a 5:1 (T cell:monocyte) ratio, after which a cytotoxicity assay against K562 cells was performed ($n = 2$; biologically independent samples). Normality was assessed using the D'Agostino & Pearson test, the Shapiro−Wilk test and visualised using a QQ plot. Box-and-whisker plots show the median (centre line), the interquartile range (25th−75th percentiles; box), and the full data range (whiskers, minimum to maximum). Parametric data are presented as mean ± SD. Non-parametric data are presented as median ± 95% CI. Data in (**A**(i-iii)) were analysed using a two-tailed, Mann−Whitney t-test. Data in (**A**(iv), **B**) were analysed using two-way ANOVA mixed effect analysis with Uncorrected Fisher's LSD. Data in (**C**) were analysed using two-way ANOVA mixed effect analysis with Šídák multiple comparison test. Data in (**E**) were analysed using a two-tailed, parametric, paired t-test. Source data are provided as a Source Data file.

(Thermofisher, 200-15) for 4 days. CD8 T cells and monocytes were cultured at a 5:1 ratio.

K562 target cells (ATCC, CCL-243) were loaded with 15 μM Calcein-AM (Thermofisher) for 30 min at 37 °C. Cells were washed and seeded at a 20:1 effector to target ratio in a 96-well U-bottom plate. NK cells were used as a positive control (maximum Lysis). K562 cells alone were used as a negative control (spontaneous release). Co-culture plates were spun at 400 g for 1 min and subsequently incubated for 6 h at 37 °C. Following incubation, samples were spun at $400 \times g$ for 5 min and 100 μL of supernatant was transferred to a black flat-bottom 96-well plate. Fluorescence was quantified using an Infinite 200Pro plate reader (Ex: 488 nm, Em: 517 nm). Specific lysis was calculated as: (Sample − Spontaneous release)/(Maximum Lysis − Spontaneous release) × 100.

### Fluorescent activated cell sorting (FACS)
Cell sorting was performed on the BD FACSDiscover S8 cell sorter. PBMCs were isolated, enumerated and stained in 0.2 μL antibody per 1 million cells for 30 min at 4 °C. Samples were washed and resuspended in PBS prior to sorting. Sorted cells were collected into PBS supplemented with 50% FBS.

### Western blotting
Monocyte subsets were isolated using FACS. Cell pellets with lysed with RIPA buffer (Sigma-Aldrich), supplemented with phosphatase and protease inhibitor (Cell Signalling), vortexed and left on ice for 30 min. Samples were spun at $13,000 \times g$ for 20 min at 4 °C. The supernatant was combined with reducing agents (Life Technologies) and denatured for 5 min at 95 °C. Extracts were separated by protein electrophoresis using a 10% Bis-Tris pre-cast gel (NuPage) and transferred overnight at 4 °C onto Hybond PVDF membrane (GE Healthcare). Membranes were blocked in ECL blocking reagent (GE Healthcare) for 1 h prior to probing with phospho-p38 MAPK (T180/Y182; 9211) overnight at 4 °C. The membrane was washed and incubated with HRP-conjugated secondary antibodies (GE Healthcare, 1:4000 dilution) for 1 h at room temperature. Antibodies were detected using the ECL detection kit (GE Healthcare).

### Statistical analysis
All statistical analysis was carried out using GraphPad Prism. Normalisation tests to determine normal or lognormal distribution were carried out on each data set. Normalisation tests used were the D'Agostino and Pearson test and the Shapiro−Wilk test and data were visualised using a QQ plot. Data sets with a normal distribution are presented as mean ± standard deviation (SD). Non-parametric data are presented as median ± 95% confidence interval (CI). During the course of the human study, there were instances where blood collection was not possible due to participant refusal. Furthermore, some blister

samples were unable to be analysed due to technical issues. Therefore, the n numbers between study groups and timepoints vary. For peripheral blood analysis $n$ numbers are between 8−12 in each group. For blister cell analysis $n$ numbers are between 5−12 in each group. For statistical analysis where there are two dependent variables, in this case timepoint and treatment, and one independent variable (e.g. cell count or lipid concentration) we used a two-way ANOVA mixed effect model. A large proportion of the data presented here is lognormally distributed, rather than normally distributed. In this case, we decided to proceed with a two-way ANOVA mixed-effect model (to account for missing values). In all other cases, where data were normally distributed we used two-way ANOVA, one-way ANOVA or a parametric t-test, as appropriate. With non-parametric data we used a Kruskal−Wallis test or Mann−Whitney non-parametric t-test. A $p$ value below 0.05 was considered significant.

### Reporting summary
Further information on research design is available in the Nature Portfolio Reporting Summary linked to this article.

## Data availability
Data supporting the findings of this study are available within the paper and its supplementary information. The raw LCMS files, protocols and metadata were uploaded to MassIVE (https://massive.ucsd.edu/) under the accession numbers MSV000099360 [https://massive.ucsd.edu/ProteoSAFe/dataset.jsp?task=95ad1cefa06a444b82fe9ef249cae4ff] and MSV000099361 [https://massive.ucsd.edu/ProteoSAFe/dataset.jsp?task=1bd3d1a3933b41898cda37739ca9c51e]. Raw FCS files and IHC files are available upon request. Source data are provided with this paper.

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

## Acknowledgements

We would like to acknowledge the financial support of the Arthritis UK (21920) studentship award to Bracken/Gilroy. This work was also supported by the Biotechnology and Biological Sciences Research Council (BBSRC) research grant (BB/X016854/1 to D.W.G) This work was supported in part by the Division of Intramural Research, National Institute of Environmental Health Sciences (NIEHS), NIH (Z01 ES025034 to D.C.Z). This work was also supported by the Medical Research Council (MRC) Grand Challenge in Experimental Medicine (MICA) Grant (MR/M003833/1 to A.N.A) and the Biotechnology and Biological Sciences Research Council (BBSRC) research grant (BB/Y003365/1 to A.N.A).

## Author contributions

Conceptualisation: DWG, DBB Methodology: O.V.B., P.J., J.W.R.G., L.B., M.L.E., F.B.L., E.S.C., H.T., M.M., K.T.F., R.D.M., J.G.E. and J.S.C. Investigation: O.V.B., E.S.C. and M.L.E. Visualisation: O.V.B., M.L.E. and E.S.C. Funding acquisition: D.G., D.B.B., A.N.A. and D.C.Z. Project administration: D.G. Supervision: D.G., D.B.B. and A.N.A. Writing—original draft: D.W.G. and O.V.B. Writing—review and editing: D.W.G., O.V.B., D.B.B. and E.S.C.

## Competing interests

O.V.B. and D.W.G. have filed a patent application for submission to the US Patent Office for the use of GSK2256294 in the treatment of chronic inflammatory disease (60-320). The remaining authors declare no competing interests.
