## [Peer Review File · Nature Communications]

EPOXY OXYLIPINS DIRECT MONOCYTE FATE IN INFLAMMATORY RESOLUTION IN HUMANS

Corresponding Author: Professor Derek Gilroy

Version 0:

Reviewer comments:

Reviewer #1

(Remarks to the Author)

This manuscript investigates the effect of acute inflammation on epoxy-oxylipin metabolism and describes a novel finding that these epoxy-oxylipins could potentially modulate monocyte differentiation in humans. The findings could have a significant impact on the field of oxylipins and the inflammation process. This study represents the first human study that reveals how acute inflammation alters the epoxy-oxylipin metabolic enzyme expression and associated metabolite levels in both plasma (systemic) and inflamed tissues. In addition, because multiple clinical candidates of soluble epoxide hydrolase inhibitors, including the one used in this study, have been developed and the target diseases are highly related to inflammation, this study may shine a light on the molecular mechanism of how the inhibitors work which remains understudied. Besides, this study represents the first study that connects epoxy-oxylipin to monocyte differentiation. All of these represent a novel, original study. However, there are several key weaknesses in experimental design, statistical analysis, and discussions that require corrections as follows:

- 1) The author claimed that inhibition of sEH preferentially elevated 12,13-EpOME and 14,15-EET only; however, the oxylipin analytical method used in the study does not cover all epoxy-oxylipin and corresponding dihydroxy-oxylipin.
 - 2) The lipidomic analysis method was under-described, there are a lot of missing details that prohibit one from evaluating if the method is appropriate, for example, what is the limit of detection/ quantifications of each metabolite, what are the instruments that were used in this method, was t-AUCB added after the blood was collected or after plasma separation? How was the t-AUCB added? Was antioxidant added for the analysis? These lipids are highly sensitive to oxidation. What is the linear range of the analysis?
 - 3) The author should, at the very least, include the 12,13-EpOME level in Fig 1 to discuss how an elevation of CYP450 and sEH modulates the epoxy and dihydroxy-oxylipins.
 - 4) When referring to the data in the discussion, the author should reference the figures to make it easier for the reader to read.
 - 5) There are multiple occasions in which the author claimed inflammation increases the epoxy-oxylipin's levels without the actual data.
 - 6) The authors also suggested that blocking inflammation-elevated epoxy-oxylipins by sEH inhibition is beneficial, but the outcome of sEH inhibition should increase / at least maintain epoxy-oxylipins. Therefore, the discussion is contradict to the main theme of this study.
 - 7) One of the key findings of this study is the discovery of epoxy-oxylipin's effects on monocyte differentiation, affecting inflammation. The author suggests that sEH inhibitors reduce intermediate monocytes and increase T cell death. However, at most, there was a couple of % increase in T cell death with large variability.
 - 8) The author has repeatedly stated that the sEH inhibitor treatment alleviated UV-KEc-induced effects. However, statistical analyses between the treatment and control groups are lacking.
 - 9) Most experimental procedures in the method section are very brief. The author is strongly encouraged to provide more details to help others to reproduce the data.
- The following are some minor comments that could improve this manuscript.
- 1) Several new clinical studies suggest that sEH inhibitor treatment is safe. The author should cite the updated references.
 - 2) The author should include a detailed breakdown regarding the population of the recruited patients in this study.

Reviewer #2

(Remarks to the Author)

This study examines the effects of the selective sEH inhibitor GSK2256294 on inflammation using a transient skin inflammation model in humans. The analysis of lipid products in blood and skin provides valuable insights into real-time changes during localized inflammation. However, the authors attempt to infer new functional roles for intermediate monocytes, which the current data do not support. As a result, the paper's conclusions are overstated and not fully backed by the experiments conducted.

Major Concerns: The flow cytometric analysis of monocytes in Figures 4 and 6 lacks clarity and supporting data. The authors should:

1. Clearly explain the rationale for investigating blood monocytes in Figure 4, following findings from Figures 1–3, and the experimental approach. This is not explained in this part of the results section.
2. Provide flow cytometric plots to illustrate gating strategies and confirm robustness of the UMAP. Also to provide context to the populations gated on for Figures 4C and D.
3. Verify that Figure 4B is correctly labeled, as the CM-IM population appears inconsistent with the UMAP data. CM-IM look more like CM(1) than CM (2) while the CM(2) population is integrated into the CM(1) population in the UMAP.
4. Justify the choice of cell identity assignments and antibody panel, as markers like CD206, CD163, and CD64 are more typical of differentiated macrophages than blood monocytes.
5. Improve clarity in Figure 4D, where the suggested changes to IM are not clear, and seem to be more prevalent at 4 hours than 24—quantification would help.
6. Clarify why the UMAP in Figure 4E (referred to as 4F in the text) has a different shape.
7. Include flow cytometric plots for Figure 6 to confirm cell gating and demonstrate that there are sufficient cell numbers per gate. The complete separation of CM and IM populations in Figure 6's UMAP is inconsistent with the transition from CM to IM shown in Figure 4 and needs explanation.
7. Standardize cluster labeling between Figures 4 and 6—e.g., why are CD205+ cells considered moDCs in skin but CM in blood?

Additional Comments:

1. Figure 5 requires flow cytometry data to validate the CM-to-NCM differentiation model and demonstrate that differentiation of CM occurs in vitro.
2. The claim that in vitro conversion to IM is blocked (Figure 5A) is overstated—there is only a slight decrease in IM numbers.
3. Figure 5B labeling should be consistent with the rest of the paper (e.g., CM instead of CD14+).
4. Representative histograms would strengthen Figure 5C, where gMFI changes are minimal.
5. The increase/stabilization of NCMs with Losmapimod in Figure 5E is striking but unexplained. How does this fit with the decrease in IMs?
6. The conclusions from Figure 7 are speculative. For example the statement that "Neutrophil numbers were unaltered with the reduction in tissue intermediate monocyte-derived macrophages indicating that the removal of neutrophils was carried out by classical monocyte-derived macrophages earlier in the response" is not supported by the data. The authors have also not referred to monocyte-derived macrophages before this section. Other experiments are needed to demonstrate this interaction.
7. More experiments are needed to directly demonstrate that IMs support CD4 T cells in the skin. This is derived from a trend towards a decrease in CD4 T cells in the context of fewer IMs which is not statistically significant, and the authors cannot infer a causal link between the 2 observations. The statement "This was confirmed by showing that intermediate monocytes directly activate CD4 T cells in the absence of any co-stimulation" does not support the conclusion.
8. The discussion over interprets the data and is very long, particularly in linking findings to opioid use—this section should be removed.
9. Figures need better organization: e.g. graph sizes are inconsistent, axis labels are too small to read clearly, p-value notations vary (use of "ns" or the exact value is given), and legends lack clarity on n values and replicates.

Version 1:

Reviewer comments:

Reviewer #1

(Remarks to the Author)

This manuscript is the first report of investigating clinical candidate of sEH inhibitor GSK2256294 effect on acute inflammation in human patient. In addition, this study demonstrated that sEH inhibitor modulates intermediate monocyte population, having a huge implication of chronic inflammatory diseases and the mechanism of the effect of sEH inhibition in different diseases. Given that sEH has beneficial effects in multiple inflammatory disease models, this studies demonstrated the potential translatability of sEH inhibitors in human diseases. The studies are clearly described and easy to follow. The authors did an excellent job to address previous concern. There are still some minor concerns as below.

- 1) on Page 7, 12,13-EpOME is not sEH metabolites. Also, 17,18-DiHETE is sEH metabolites but not CYP450 metabolites.
- 2) The figure 5Bi needs a proper standard size markers. It is also recommended to submit all western blot pictures.
- 3) The caption of Figure 5C is not very clear.

4) The author should upload the metadata of the oxylipin analysis.

Reviewer #2

(Remarks to the Author)

The authors have worked hard to address my comments and have included a significant amount of new information which strengthens the manuscript. Whilst the data on GSK2256294 are interesting and very relevant to the management of human pain and inflammation, my concern remains that the link to a mechanistic role for intermediate monocytes cannot be made from these data without more mechanistic experiments. Because of this I find the statements in the manuscript linked to this, such as in the abstract "revealing a role for intermediate monocytes in maintaining CD4 T cell viability during inflammation-resolution" are still over-stated.

With respect to the naming of the cell clusters, I still believe that there is inconsistency between the figures that should be resolved. If, as the authors say in their response, they have selected genes in figure 4 to capture the full spectrum of monocytes/mo-macrophages that could be seen in blood and tissue, why have they selected to use a different panel including CD163 macrophages in the skin? Likewise the "CM" in Figure 6 in the skin do not express CD14 so why are they labelled as such. This is important because subjective labelling of the clusters makes it harder for the reader to make their own conclusions about the cells present in the blood versus tissue.

For Figure 7 flow cytometry data showing shared expression of the defining CD4 T cell markers would help the reader understand which populations are present. For these data I am missing a crucial control of CM + T cells. How can the authors be confident that the T cell phenotype is IM-specific rather than due to T cell interaction with any APC? This would be a vital step towards the claim that IM mechanistically alter T cell responses in the skin in the presence of GSK2256294.

Version 2:

Reviewer comments:

Reviewer #2

(Remarks to the Author)

The authors have addressed my comments and I appreciate the addition of the new supplementary figure 18 which is useful in understanding the spread of gene expression between blood and skin monocytes.

Since the co-culture data do not show a clear difference between classical and intermediate monocytes (and, in several cases, non-classical monocytes) in their effect on T cell phenotype, I maintain that the claim that intermediate monocytes specifically support CD4⁺ T cell viability can be discussed in the manuscript but omitted from the abstract.

The information provided in the rebuttal would be a useful addition to the discussion: "However, given that IMs are significantly higher in number compared to CM (Figure 7c) – 350 CM versus 1800 IMs – and that only IMs are reduced in the skin following GSK2256294 coincident with reduced T cells (Figure 7A) we logically concluded that during resolving inflammation when IM predominate, they maintain CD4 T cell viability."

I suggest that the paper and the abstract contain enough interesting information on the impact on monocytes without the need to infer a T cell mechanism that hasn't been proved.

REVIEWER 1

This manuscript investigates the effect of acute inflammation on epoxy-oxylipin metabolism and describes a novel finding that these epoxy-oxylipins could potentially modulate monocyte differentiation in humans. The findings could have a significant impact on the field of oxylipins and the inflammation process. This study represents the first human study that reveals how acute inflammation alters the epoxy-oxylipin metabolic enzyme expression and associated metabolite levels in both plasma (systemic) and inflamed tissues. In addition, because multiple clinical candidates of soluble epoxide hydrolase inhibitors, including the one used in this study, have been developed and the target diseases are highly related to inflammation, this study may shine a light on the molecular mechanism of how the inhibitors work which remains understudied. Besides, this study represents the first study that connects epoxy-oxylipin to monocyte differentiation. All of these represent a novel, original study. However, there are several key weaknesses in experimental design, statistical analysis, and discussions that require corrections as follows:

Comment 1. The author claimed that inhibition of sEH preferentially elevated 12,13-EpOME and 14,15-EET only; however, the oxylipin analytical method used in the study does not cover all epoxy-oxylipin and corresponding dihydroxy-oxylipin.

Author response: A comprehensive list of major metabolites of PUFAs by CYP450s following prophylactic and therapeutic studies is provided in supplementary figure 10.

Comment 2. The lipidomic analysis method was under-described, there are a lot of missing details that prohibit one from evaluating if the method is appropriate, for example, what is the limit of detection/ quantifications of each metabolite, what are the instruments that were used in this method, was t-AUCB added after the blood was collected or after plasma separation? How was the t-AUCB was added? Was antioxidant added for the analysis? These lipids are highly sensitive to oxidation. What is the linear range of the analysis?

Author response: We apologize for the brief description of the lipidomic methods and reliance on previous references. We have expanded the description in methods as cited below and added 2 supplemental figures to address these issues (Supplementary Figure 16 and Supplementary Table 4). The table includes transitions, retention times, ISTDs employed as well as the low and high limit of the standard curves (in pg/ul injected). These standard curves were all linear within the range, as evidenced by visual inspection and indicated by the high r^2 values. All the minimum standards serve as LLOQ as all these peaks for these analytes exceed $S/N > 10$. Supplementary figure 10 shows values for all analytes where most or all samples had peak responses with $S/N > 10$. tAUCB was added by the Zeldin lab during or after thawing immediately prior to extraction. We observe substantial oxidation of fatty acids and oxylipins when using SPE; however, we do not have similar issues with liquid:liquid extractions. In the past we have seen no benefit of the addition of antioxidants in this assay and none were used in this study.

“Plasma samples (200 ul) were thawed and diluted 1:1 with (5 % methanol and 0.1 % acetic acid). To prevent epoxy-oxylipin catabolism by sEH during extraction, 1 ul of t-AUCB in methanol (sEH-I; 10 μ M final concentration) was added to each sample.

After addition of internal standards (11,12-DHET-d11 (150ng), 11,12-EET-d11 (300ng), 15-HETE-d8 (300ng)), extraction was performed by liquid:liquid extraction using 1 ml ethyl acetate. The samples were dried and reconstituted in 50 μ l of 30 % glycerol. 10 μ l injections were assayed on an Ultimate 3000 UHPLC equipped with an Xselect CSH C18, 2.1 \times 50 mm, 3.5 μ m particle column (Waters, Wilford, MA) and a TSQ Quantiva tandem mass spectrometer (Thermo Fisher Scientific, Waltham, MA) as previously described⁶¹. Relative response ratios of analytes were compared to linear standard curves generated with purchased oxylipins (Cayman Chemical, Ann Arbor, MI). Analyte retention times, ionizations, internal standards used and the range and r² of the standard curves is shown in Supplemental Table 3. Representative chromatograms for 14,15-EET, 14,15-DHET, 12,13-EpOME and 12,13-DiHOME from a plasma sample are presented in Supplementary Figure 16”

Comment 3. The author should, at the very least, include the 12,13-EpOME level in Fig 1 to discuss how an elevation of CYP450 and sEH modulates the epoxy and dihydroxy-oxylipins.

Author response: Included as suggested in Figure 1D vi.

Comment 4. When referring to the data in the discussion, the author should reference the figures to make it easier for the reader to read.

Author response: We have made these additions, but, as this is unusual practice, we leave the final decision to the editor.

Comment 5. There are multiple occasions in which the author claimed inflammation increases the epoxy-oxylipin’s levels without the actual data.

Author response: See figure 1, panel D (vi and vii) where 12,13-EpOME and total products, for instance, are elevated in the inflammatory exudates after intradermal injection of UVKEc in humans.

Comment 6. The authors also suggested that blocking inflammation-elevated epoxy-oxylipins by sEH inhibition is beneficial, but the outcome of sEH inhibition should increase / at least maintain epoxy-oxylipins. Therefore, the discussion is contradicted to the main theme of this study.

Author response: We fear that there is some confusion on this point. To clarify our data, we indeed show that sEH inhibition elevates epoxy oxylipins above untreated control, see Figures 3 and 6, an expected outcome. This is associated with greater resolution of pain and a block in the expansion of intermediate monocytes, the implication of which, in chronic inflammation, are discussed.

Comment 7. One of the key findings of this study is the discovery of epoxy-oxylipin’s effects on monocyte differentiation, affecting inflammation. The author suggests that sEH inhibitors reduce intermediate monocytes and increase T cell death. However, at most, there was a couple of % increase in T cell death with large variability.

Author response: sEH inhibition caused an increase in T cell death from 1.5% to 5.1%, with a variable SD as noted. This, in turn resulted in a reduction in CD4 T cells

from 672 cells/blister to 105 cells/blister at the site of inflammation, see **Figure 7A**. The question is – to what extent could this increase in percentage of cell death account for the corresponding reduced cell numbers. As stated by this reviewer, variability is an issue. But unlike rodents, human inflammation biology can be very variable in its responses.

Now, we can certainly play with a hypothetical number pertinent to the current scenario: assume we have an initial number of CD4 T cells of 600 with a doubling time of 72 hours, a concomitant death rate of 5%/hour (i.e., survival = 95% per hour) over 24h. This would result in approximately 221 cells remaining after 24h.

However, with limited information this is only conjecture. We are proposing that intermediate monocyte-derived macrophages keep tissue CD4 T cells alive, which is supported by the outcome of *in vitro* co-culture experiments presented in **Figure 7E**. However, there are inherent difficulties in predicting the precise extent to which this happens *in vivo*. Specifically, the impact of reducing the presence of a viability factor (intermediate monocyte-derived macrophages in this case) on numbers of surviving cells in a complex tissue microenvironment as we don't know (1) the precise rate of CD4 T cell proliferation, (2) the rate of cell death, (3) ongoing infiltration alongside (4) efflux to draining lymph nodes.

Therefore, we reasonably suggest stating that “*However, numbers of CD4 T cells were reduced at 48 hrs following GSK2256294 coincident with an increase in T cells acquiring a live dead stain and remnants of immune debris. These findings are consistent with intermediate monocytes being able to maintain T cell viability creating a tentative link with helping to imprint long-term tissue immunity following infection*”. This can be found in line 378 of the manuscript.

Comment 8. The author has repeatedly stated that the sEH inhibitor treatment alleviated UV-KEc-induced effects. However, statistical analyses between the treatment and control groups are lacking.

Author response. These analyses have now been included in Figure 3, Figure 4 and Figure 6 where we have included comparisons between the drug and control groups for lipidomic analysis, IMs at 24 hrs in peripheral blood and IMs at 24 hrs in tissue

Comment 9. Most experimental procedures in the method section are very brief. The author is strongly encouraged to provide more details to help others to reproduce the data.

Author response. These have been expanded as suggested including western blotting, PBMC isolation, flow cytometry, immunohistochemistry and an expanded on lipidomics analysis.

Minor comments

Comment 1. Several new clinical studies suggest that sEH inhibitor treatment is safe. The author should cite the updated references.

Author response: We have included references in the paper (page 14, line 319) that confirm the safety of sEH inhibitors for use in humans.

Comment 2. The author should include a detailed breakdown regarding the population of the recruited patients in this study.

Author response: We included a table of characteristics on page 19, line 442, but as this is not a clinical trial we didn't record weight/height/ethnicity.

REVIEWER 2

This study examines the effects of the selective sEH inhibitor GSK2256294 on inflammation using a transient skin inflammation model in humans. The analysis of lipid products in blood and skin provides valuable insights into real-time changes during localized inflammation. However, the authors attempt to infer new functional roles for intermediate monocytes, which the current data do not support. As a result, the paper's conclusions are overstated and not fully backed by the experiments conducted.

Comment: Major Concerns: The flow cytometric analysis of monocytes in Figures 4 and 6 lacks clarity and supporting data. The authors should:

Author response: We have addressed these flow cytometric issues in points 2-9 below.

Comment 1. Clearly explain the rationale for investigating blood monocytes in Figure 4, following findings from Figures 1–3, and the experimental approach. This is not explained in this part of the results section.

Author response: When conducting interventional experiments for the first time especially in humans (it's easy to re-order more mice from Charles River etc, but recruiting humans is less easy), an exploratory approach is best – shake the tree and see what falls out, as it were, and design your experiments to capture this knowledge based on extensive experience. A hypothesis driven approach runs the risk of being too prescriptive and missing something fundamentally important. This is precisely the approach we took to these studies. Certainly, we had prior knowledge of the model (PMID: 27499924) and data showing the CYP450 products are elevated in tissues 14hrs after UVKEc injection (Figure 1 vii) mirroring mononuclear

phagocyte infiltration. Explaining why a particular trail of investigation was followed, will only lengthen an already long manuscript and add little to the overall take home point.

Comment 2. Provide flow cytometric plots to illustrate gating strategies and confirm robustness of the UMAP. Also to provide context to the populations gated on for Figures 4C and D.

Author response: Gating strategies have been included in Supplementary Figure 11 to show the gating strategies for the manual gating in Figure 4A and to show which population was used in the generation of the UMAPs in 4B.

Comment 3. Verify that Figure 4B is correctly labelled, as the CM-IM population appears inconsistent with the UMAP data. CM-IM look more like CM(1) than CM (2) while the CM(2) population is integrated into the CM(1) population in the UMAP.

Author response: We confirm that the UMAP has been labelled correctly as the CM-IM monocyte population is gaining the expression of CD16, which is mirrored in the manual gating.

Comment 4. Justify the choice of cell identity assignments and antibody panel, as markers like CD206, CD163, and CD64 are more typical of differentiated macrophages than blood monocytes.

Author response: In this study we are dealing with two compartments - peripheral blood and inflamed tissue with cells migrating from blood into tissue in a dynamic continuum. Hence, for direct comparison purposes, we designed the panel of archetypical markers to capture mononuclear phagocytes in both compartments bearing in mind that some blood monocytes also share markers with tissue macrophages. Therefore, it was important to include both monocyte markers and markers more typical of tissue macrophages with CD64, for instance, being an important monocyte marker, as well as being expressed on macrophages. These markers also inform on the phenotype of tissue mononuclear phagocytes.

Comment 5. Improve clarity in Figure 4D, where the suggested changes to IM are not clear, and seem to be more prevalent at 4 hours than 24—quantification would help.

Author response: We have updated this figure and have quantified the UMAP population of IMs in Figure 4B.

Comment 6. Clarify why the UMAP in Figure 4E (referred to as 4F in the text) has a different shape.

Author response: Monocyte marker expression changes when monocytes enter the tissue, hence the change in shape of the UMAP from peripheral blood monocytes to mononuclear phagocytes that have been isolated from inflamed tissue. Below, we have provided graphs that show how the MFIs of the markers we have used differ between blood monocytes at 24 hrs and tissue monocytes at 24 hrs for the reviewers in the figure below:

Monocyte phenotype in blood vs. blister 24 hrs after inflammatory onset.

The forearms of participants were intradermally injected with UV-killed *E. coli* (UVKEc) resulting in a local and peripheral inflammatory response. Peripheral bloods and local inflammatory exudate were collected and analysed at baseline and 4 hrs, 24 hrs and 48 hrs post UVKEc injection. Data shows monocyte expression for (A) HLA-DR, (B) CCR2, (C) CD64, (D) CD86, (E) CD14, (F) CD16, (G) CD11c and (H) CD163 on (i) peripheral blood or (ii) blister monocyte subsets at 24 hrs post UVKEc. Data are presented as mean \pm SD for $n = 11$ or $n = 6$ for both groups. All graphs were analysed using one-way-ANOVA mixed effect analysis with Sidak's multiple comparisons test, * $p < 0.05$, ** $p < 0.01$, *** $p < 0.001$, **** $p < 0.0001$.

Furthermore, we know that there is a linear differentiation pathway for classical monocytes to intermediate monocytes, and then non-classical monocytes in blood (PMID: 28606987). However, we cannot say with certainty that there is a similar linear differentiation process in inflamed tissues, or whether respective blood monocyte populations migrate into inflamed tissue in a sequential manner. The latter is supported by the fact that sEH inhibition reduced numbers of blood intermediate monocytes (Figure 4), and consequently fewer of these cells were found in the

inflamed skin, Figure 6. Indeed, given the differing shape of the UMAP we would hypothesise that differentiation is not occurring in the tissue.

Comment 7. Include flow cytometric plots for Figure 6 to confirm cell gating and demonstrate that there are sufficient cell numbers per gate. The complete separation of CM and IM populations in Figure 6's UMAP is inconsistent with the transition from CM to IM shown in Figure 4 and needs explanation.

Author response: Flow cytometry plots representative of a 24h blister sample have been included in Supplementary Figure 11. There are several reasons for the differences between UMAPs in Figure 4 versus Figure 6. In peripheral blood, classical monocytes are the predominant population making up over 80% of the total monocyte population followed by 10% each of intermediate and non-classical. However, in the inflamed skin, mononuclear phagocytes bearing an intermediate phenotype are quantitatively the predominate cells type (Figure 7C), arising from infiltrating blood intermediate monocytes as argued above. Moreover, in blood, monocytes are in a relatively low state of activation which changes dramatically when they transmigrate across the microvascular endothelium and into inflamed tissue niches. Therefore, this difference in UMAP between Figure 4 and Figure 6 is a facet of relative proportions, quantities and activation status.

We have clarified in the text that we expect that linear differentiation is not occurring in tissue, see page 15 line 333.

Comment 8. Standardize cluster labeling between Figures 4 and 6—e.g., why are CD205+ cells considered moDCs in skin but CM in blood?

Author response: In humans, classic monocytes are identified using standard markers including HLA-DR⁻, CD14⁺ and CD16⁻. As stated above, we used the same markers for blood monocytes as for tissue mononuclear phagocytes. Given that we would not expect to see monocyte-derived DCs in blood, CD205 is simply higher in blood classical monocytes as is consistent with the literature. In contrast, in the inflamed tissue, mononuclear phagocytes positive for CD205 are likely to be monocyte-derived DCs. Hence, the differences between blood monocytes and tissue mononuclear phagocytes are due to monocytes acquiring a new phenotype once they arrive at inflamed tissue niches having migrated across the microvascular endothelium, as described above.

In tissue there will be a multitude of inflammatory cues that result in the differentiation of monocytes into different macrophage populations and monocyte derived DC populations. These signals will be absent from blood.

Additional comments

Comment 1. Figure 5 requires flow cytometry data to validate the CM-to-NCM differentiation model and demonstrate that differentiation of CM occurs in vitro.

Author response: In **Supplementary Figure 14** we have included gating strategies for monocytes from freshly isolated PBMCs and PBMCs that have been cultured for 24 hrs to show the transition from classical monocytes into intermediate monocytes

Comment 2. The claim that *in vitro* conversion to IM is blocked (Figure 5A) is overstated—there is only a slight decrease in IM numbers.

Author response: The mean percent of intermediate monocytes of total monocytes decreases from 53% to 41% with addition of 12,13-EpOME, which we appreciate is not a total block and we have amended the language in the paper accordingly (line 238). We would point out that these lipids are biologically unstable and it is likely that their stability will be reduced *in vitro* compared to the continuous synthesis that occurs *in vivo*, especially given that this was a single dose of lipids over a 24h period. *In vivo* this is a more dynamic process where cells are continuously exposed to constitutively high level of lipids, result in the effect being more prolonged and potent

Comment 3. Figure 5B labelling should be consistent with the rest of the paper (e.g., CM instead of CD14+).

Author response: To not tamper with the original western blot image we did not change the labelling of the monocyte subsets, however, if the editors wish us to do so then we can.

Comment 4. Representative histograms would strengthen Figure 5C, where gMFI changes are minimal.

Author response: We have added in a histogram in figure 5C, showing the shift to the left of the curve with the addition of 12,13-EpOME. We appreciate the shift is subtle but is consistent across 10 experimental repeats.

Comment 5. The increase/stabilization of NCMs with losmapimod in Figure 5E is striking but unexplained. How does this fit with the decrease in IMs?

Author response: This is likely a facet of the relative abundance of classical monocytes in resting blood of humans (80%) to that of intermediate monocytes (10%) as well as the rate of differentiation of classical monocytes to intermediate monocytes under steady data, which data in Figure 5E represents.

During steady state, 1% of the abundant classical monocytes differentiate into less abundant intermediate monocytes within 24h. In contrast, intermediate monocytes persist in blood for 4.3 days while non-classical monocytes persist in blood for 7 days, (PMID: 28606987). As losmapimod was given to participants for four days without inflammation and, in keeping with the dynamic liner differentiation of classical to intermediate monocytes during steady state, it would be unlikely to see a significant reduction in non-classical monocytes during this time.

Comment 6. The conclusions from Figure 7 are speculative. For example the statement that “Neutrophil numbers were unaltered with the reduction in tissue intermediate monocyte-derived macrophages indicating that the removal of neutrophils was carried out by classical monocyte-derived macrophages earlier in the response” is not supported by the data. The authors have also not referred to

monocyte-derived macrophages before this section. Other experiments are needed to demonstrate this interaction.

Author response: We agree, and we have toned down this assertion, page 16 line 376.

Comment 7. More experiments are needed to directly demonstrate that IMs support CD4 T cells in the skin. This is derived from a trend towards a decrease in CD4 T cells in the context of fewer IMs which is not statistically significant, and the authors cannot infer a causal link between the 2 observations. The statement “This was confirmed by showing that intermediate monocytes directly activate CD4 T cells in the absence of any co-stimulation” does not support the conclusion.

Author response: See response to Reviewer 1, point 7. Also in discussion, page 16 line 378.

Comment 8. The discussion over interprets the data and is very long, particularly in linking findings to opioid use—this section should be removed.

Author response: Agreed! The discussion has been shortened.

Comment 9. Figures need better organization: e.g. graph sizes are inconsistent, axis labels are too small to read clearly, p-value notations vary (use of “ns” or the exact value is given), and legends lack clarity on n values and replicates.

Author response: All have been corrected line with suggestions made.

Reviewer #1 (Remarks to the Author):

This manuscript is the first report of investigating clinical candidate of sEH inhibitor GSK2256294 effect on acute inflammation in human patient. In addition, this study demonstrated that sEH inhibitor modulates intermediate monocyte population, having a huge implication of chronic inflammatory diseases and the mechanism of the effect of sEH inhibition in different diseases. Given that sEH has beneficial effects in multiple inflammatory disease models, this studies demonstrated the potential translatability of sEH inhibitors in human diseases. The studies are clearly described and easy to follow. The authors did an excellent job to address previous concern. There are still some minor concerns as below.

Comment 1. on Page 7, 12,13-EpOME is not sEH metabolites. Also, 17,18-DiHETE is sEH metabolites but not CYP450 metabolites.

Author response: We thank the reviewers for pointing this out. We have amended the language on p7 line 145 to clarify that 12,13-EpOME is a CYP450 metabolite. Furthermore, we have amended the language for 17,18-DiHETE on page 7 line 150.

Comment 2. The figure 5Bi needs a proper standard size markers. It is also recommended to submit all western blot pictures.

Author response: We have uploaded all the western blot images for the four donors. Exposure of the western blot was carried out using x ray film, as such, the ladder does not show up. The ladder was marked in situ using pen and is clearly labelled on the left hand side. We have included pictures of all western blots for P-p38 and the GAPDH control in **supplementary figure 19**.

Comment 3. The caption of Figure 5C is not very clear.

Author response: We have amended the caption of figure 5C to be clearer:

‘PBMCs (2×10^6 /well) were treated with 1 μ M 12,13-EpOME or vehicle (MA) for 24 h, then re-stimulated with 12,13-EpOME or MA plus 100 ng/ml LPS for 30 min. Cells were fixed and stained for HLA-DR and phosphorylated p38 (P-p38). (i) Phosphorylation of p38 in PBMCs treated with 12,13-EpOME or MA following LPS stimulation (representative of 10 experiments). (ii) Representative histogram of P-p38 MFI in PBMCs treated with LPS + MA, LPS + 12,13-EpOME, LPS + losmapimod (Los), or unstimulated.’

We hope that this is clearer for the reader.

Comment 4. 4) The author should upload the metadata of the oxylipin analysis.

Author response: The raw LCMS files, protocols and metadata were uploaded to MassIVE (<https://massive.ucsd.edu/>). The accession numbers are MSV000099360 and MSV000099361.

Reviewer #2 (Remarks to the Author):

Comment 1. The authors have worked hard to address my comments and have included a significant amount of new information which strengthens the manuscript. Whilst the data on GSK2256294 are interesting and very relevant to the management of human pain and inflammation, my concern remains that the link to a mechanistic role for intermediate monocytes cannot be made from these data without more mechanistic experiments. Because of this I find the statements in the manuscript linked to this, such as in the abstract "revealing a role for intermediate monocytes in maintaining CD4 T cell viability during inflammation-resolution" are still over-stated.

Author response: We thank the reviewer for this comment and understand their concerns. We have further toned down the language in the abstract on page 2, line 39. We have also put the new sentence below:

'Fewer intermediate-like monocytes were observed at the site of inflammation, accompanied by reduced tissue CD4 T cells, suggesting that intermediate monocytes may contribute to supporting CD4 T cell viability during inflammation resolution.'

We ask the review to look at our response to comment 3 for more information.

Comment 2: With respect to the naming of the cell clusters, I still believe that there is inconsistency between the figures that should be resolved. If, as the authors say in their response, they have selected genes in figure 4 to capture the full spectrum of monocytes/macrophages that could be seen in blood and tissue, why have they selected to use a different panel including CD163 macrophages in the skin? Likewise the "CM" in Figure 6 in the skin do not express CD14 so why are they labelled as such. This is important because subjective labelling of the clusters makes it harder for the reader to make their own conclusions about the cells present in the blood versus tissue.

Author response: We appreciate these comments. The panel that has been used to identify the clusters in both figure 4 and figure 6 is identical, as can be seen in the heatmap in both figures. The markers that have been used for both are HLA-DR, CD16, CD14, CD206, CD205, CCR2, CD64, Tim-4, CD11c, CD86 and CD163. The markers appear in a different order on each heatmap due to the differential hierarchical clustering. The list of antibodies used in clustering has also been added to the figure legends in Figure 4 and Figure 6.

To the reviewer's second concern regarding CM in the skin not expressing CD14, we have chosen to represent the marker expression in the clusters as a heatmap, which means that the expression of each marker is a scaled expression. Although there is CD14 expression on the CM, it is significantly lower than that on the IM and CM-IM (hence a darker colour on the scaled expression). For the avoidance of any doubt, we have included in **supplementary figure 18** box plots of the marker expression from the clusters in Figure 6 from the supervised clustering and

we have included the marker expression of HLA-DR, CCR2, CD64, CD86, CD14, CD16, CD11c and CD163 in each of the monocyte subsets from manual gating analysis. Both of these demonstrate that, in the blister, CD14 expression is significantly higher in the intermediate monocytes compared to CM. This differs from peripheral blood, where CD14 expression is equal between CM and IM. We hope that the inclusion of these data will also allow the reader to make their own conclusions on the clusters and clarify the significant changes to each monocyte marker upon entering the tissue.

Comment 3. For Figure 7 flow cytometry data showing shared expression of the defining CD4 T cell markers would help the reader understand which populations are present.

Author response: The gating strategy for each population has been shown in **Supplementary Figure 21** and this has been added to the manuscript on p13, line 303.

For these data I am missing a crucial control of CM + T cells. How can the authors be confident that the T cell phenotype is IM-specific rather than due to T cell interaction with any APC? This would be a vital step towards the claim that IM mechanistically alter T cell responses in the skin in the presence of GSK2256294.

Author response:

We did these experiments and found that CMs exert similar effects on CD4 T cells as IMs and we have included this data in **Supplementary Figure 20**. However, given that IMs are significantly higher in number compared to CM (Figure 7c) – 350 CM versus 1800 IMs – and that only IMs are reduced in the skin following GSK2256294 coincident with reduced T cells (Figure 7A) we logically concluded that during resolving inflammation when IM predominate, they maintain CD4 T cell viability. We have, therefore, altered the abstract and discussion to reflect a more cautious interpretation of these data. We have added a sentence for this additional supplementary information on page 13 line 304.

REVIEWERS' COMMENTS

Reviewer #2 (Remarks to the Author):

The authors have addressed my comments and I appreciate the addition of the new supplementary figure 18 which is useful in understanding the spread of gene expression between blood and skin monocytes.

Since the co-culture data do not show a clear difference between classical and intermediate monocytes (and, in several cases, non-classical monocytes) in their effect on T cell phenotype, I maintain that the claim that intermediate monocytes specifically support CD4⁺ T cell viability can be discussed in the manuscript but omitted from the abstract.

Author Response: We have altered the abstract and removed the claim that IMs specifically support CD4 T cell viability, **line 39**.

The information provided in the rebuttal would be a useful addition to the discussion: "However, given that IMs are significantly higher in number compared to CM (Figure 7c) – 350 CM versus 1800 IMs – and that only IMs are reduced in the skin following GSK2256294 coincident with reduced T cells (Figure 7A) we logically concluded that during resolving inflammation when IM predominate, they maintain CD4 T cell viability."

I suggest that the paper and the abstract contain enough interesting information on the impact on monocytes without the need to infer a T cell mechanism that hasn't been proved.

Author response: We have altered the discussion to include the reviewer's suggestions, **line 379**.